# A Primer on the Factories of the Future

**DOI:** 10.3390/s22155834

**Published:** 2022-08-04

**Authors:** Noble Anumbe, Clint Saidy, Ramy Harik

**Affiliations:** 1Department of Mechanical Engineering, University of South Carolina, Columbia, SC 29201, USA; 2McNair Aerospace Center, University of South Carolina, Columbia, SC 29201, USA

**Keywords:** smart factory, advanced manufacturing, intelligent manufacturing, cyber manufacturing, cyber physical systems, Internet of Things, Industry 4.0, artificial intelligence, data driven manufacturing

## Abstract

In a dynamic and rapidly changing world, customers’ often conflicting demands have continued to evolve, outstripping the ability of the traditional factory to address modern-day production challenges. To fix these challenges, several manufacturing paradigms have been proposed. Some of these have monikers such as the smart factory, intelligent factory, digital factory, and cloud-based factory. Due to a lack of consensus on general nomenclature, the term *Factory of the Future* (or Future Factory) has been used in this paper as a collective euphemism for these paradigms. The *Factory of the Future* constitutes a creative convergence of multiple technologies, techniques, and capabilities that represent a significant change in current production capabilities, models, and practices. Using the semi-narrative research methodology in concert with the snowballing approach, the authors reviewed the open literature to understand the organizing principles behind the most common smart manufacturing paradigms with a view to developing a creative reference that articulates their shared characteristics and features under a collective lingua franca, viz., *Factory of the Future*. Serving as a review article and a reference monograph, the paper details the meanings, characteristics, technological framework, and applications of the modern factory and its various connotations. Amongst other objectives, it characterizes the next-generation factory and provides an overview of reference architectures/models that guide their structured development and deployment. Three advanced communication technologies capable of advancing the goals of the *Factory of the Future* and rapidly scaling advancements in the field are discussed. It was established that next-generation factories would be data rich environments. The realization of their ultimate value would depend on the ability of stakeholders to develop the appropriate infrastructure to extract, store, and process data to support decision making and process optimization.

## 1. Introduction

There is a burgeoning commitment by governments, industry, and academia to the digital transformation of industry with a view to attaining the Future Factory. Though it is still in its infancy, the *Factory of the Future* is one of the key constructs of Industry 4.0. In this paper, it is envisioned as a highly connected, very intelligent, and broadly digitized production facility. It represents the future state of a well-instrumented and fully connected manufacturing entity sitting atop a cyber-physical framework. It is assumed to be highly flexible and extremely adaptable to production processes that rapidly accommodate product customization. The competitive edge of the factory of the future is that it reaches beyond the bounds of a traditional factory, which focuses on the rudimentary production of physical products and extends its reach into such far-flung functions such as production planning, production scheduling, inventory management, supply chain logistics, and even product design and development, all with limited human intervention. It is imperative to understand past and present research trends to be able to fully understand and anticipate the future of factories, and thus support the formulation of strategies for the diffusion of research findings and knowledge about the *Factory of the Future*. The purpose of this review paper is to articulate and characterize this *Factory of the Future*. The paper seeks to investigate the meanings, characteristics, and technological underpinnings of the said Factory. Specifically, this research aims at answering the following research questions: RQ1: What is the *Factory of the Future* and what are its characteristics? What are the various reference architectures, including descriptions of their components, constraints, interconnections, and interactions that help define the *Factory of the Future*? What are the communication technologies that would have the most impact on the development of the *Factory of the Future*? Since the *Factory of the Future* is data-driven, what technologies and strategies exist for seamlessly connecting all assets to enable the processing of data for intelligent decision making? What are the key enabling technologies that underpin the *Factory of the Future*, and how are they currently being applied? The rest of the paper is arranged as follows: Section 2 lays a historical background for the evolution of manufacturing, beginning with the first industrial revolution when manpower was the state-of-art. It proceeds with a system maturity model (i.e., Levels of System Sovereignty) that discriminates between the different levels of factory autonomy. It wraps up with a synopsis on the concept of Industry 4.0 and ends with an association between Industry 4.0 and the *Future Factory*. Section 3 oultlines the research methodology, and Section 4 delves deeper into articulating the concept of the *Factory of the Future* and its characteristics in the context of the manufacturing ecosystem (MeS). Section 5 discusses the conceptual framework of a typical *Factory of the Future* and the reference architectures which provide, in one model, vetted and recommended nomenclatures, structures, and integrations of various aspects (IT, OT, business, etc.) of an enterprise, necessary for the development or upgrade of the said factory. Section 6 outlines three communication standards/technologies (OPC-UA, TSN, and 5G mobile network) that would help shape the future of advanced manufacturing, and Section 7 looks at practical techniques and technologies necessary for the integration of the various assets within the factory with a view to enabling intra and inter-factory communication. Section 8 is dedicated to the key enabling technologies that make the *Factory of the Future* possible. The paper is concluded with a summary and conclusion in Section 11.

## 2. A Historical Narrative: The Past, Present and Future of the Factory and Its Ecosystem

Historical context is necessary to understand the *Factory of the Future*. This section provides a historical overview.

### 2.1. The Industrial Revolutions

Historical observers have reported a series of Industrial revolutions. These “industrial revolutions” are frames of reference for the intersection of events and emergent technologies that often led to marked shifts in productivity, industry, and society. Each of the past revolutions was driven by the emergence of new technologies and systematically resulted in wholesale disruptions and concurrent transformations in industrial processes, manufacturing methodologies, business models, and the organization of capital and labor. These shifts have not only resulted in global re-organization of the means of production, but also in remarkable changes to the socio-political, cultural, and economic fortunes of nations. We are currently in the dawn of the Fourth Industrial Revolution, and each of them can be described as follows: (a) The *First Industrial Revolution* (circa 1760 to 1840): This was the advent of mechanized production using coal, which resulted in the transition from muscle power to mechanical power [1,2,3,4]. It was triggered by the invention of the stream engine, and hydropower. It led to the emergence of the railroad construction industry. The major contribution of this era was improved efficiency. (b) The *Second Industrial Revolution* (late 19th century (circa 1870) to early 20th century): The second Industrial revolution [5,6,7] emerged in part due to the arrival of electric power and the advent of the assembly line. It enabled mass production and kick-started the era of automation. (c) The *Third Industrial Revolution* (Mid (circa 1969) to Late Twentieth Century): The third Industrial revolution [8,9,10] is otherwise known as the computer or digital revolution. Electronics and information technology were key technologies of this era. The era heralded the rise of computer networks, the emergence of the Internet, and the arrival of robots. Automated production was also a product of this era, mostly facilitated by the growth of machine control and robots. (d) The *Fourth Industrial Revolution:* This era was marked by the ubiquity of physical object representation in highly interactive virtual information networks [11,12], leading up to the blurring of the boundaries between the physical and virtual worlds. This era [13] characterized a shift in reliance on the client-server model to the adoption of ubiquitous mobility that has come to catalyze the growth of smart things. Other remarkable elements of this era include the growth of exponential technologies such as digital twins [14,15,16,17,18,19,20], artificial intelligence (AI) [21,22,23,24,25,26,27,28], simulations [29,30,31,32,33,34,35], blockchain [36,37,38,39,40,41], big data and analytics [42,43,44,45,46], augmented and virtual reality (AR, VR) [47,48,49,50,51,52,53,54], and robotics [55,56,57,58,59,60,61]. Industry 4.0 is a construct of the Fourth Industrial Revolution. It seeks to bring together the various conceptual elements capable of framing the eminent transformation expected from the collision of these technologies and the events they would likely trigger. The *Future Factory* is one of many outcomes of this construct. Though industry watchers have identified and focused on the four industrial revolutions referenced above, there are early speculations about a *fifth industrial revolution*. There is currently a lack of consensus on a definition for this emerging revolution. However, current information suggests that the main distinction between the Fourth and Fifth Industrial Revolutions could surround the role of humans and the extent of their intervention in manufacturing. There are concerns that the full realization of the Fourth Industrial Revolution could alienate humans. There is therefore a strong debate being had around the idea of redefining the role of humans in the factory. This could mean bringing humans back as the central feature of the factory through a deliberate emphasis on strong human–machine interaction.

### 2.2. Levels of System Sovereignty (L2S): A System Maturity Model

To evaluate the degrees of sovereignty of systems, machines, or other industrial components for each of the four industrial periods, we propose a system maturity model referred to as “Levels of System Sovereignty (L2IS). It shows the measure, state, or degree of a system’s autonomy or its ability to self-govern, without human intervention. The model divvies up the different stages of industrial (manufacturing) development based on the degrees to which tasks, system control, and authority are split between humans and machines (or more broadly, technology).

Humans and machines have always shared control or authority in a mutually complimentary manner within the manufacturing space. What has varied is the degree of autonomy (control, authority, or sovereignty) maintained by either party. Based on this model, a factory (or any other qualifying industrial system) can be said to have at one of four (4) sovereignty levels: mechanization, automation, semi-autonomy, or autonomy (Near complete or full autonomy). Given this classification, the higher the autonomy of a system, the lower the demand for human-level intelligence or manual intervention. The model is further discussed using Figure 1.

#### 2.2.1. Mechanization

At this stage of industrialization, machines (often heavy industrial machinery) were used to partially or completely do the work that was previously done using manual labor. Manual labor here refers to work performed by humans without any tools or support. Mechanization was a distinct feature of the first and second industrial revolutions. Though machines became the work horses of the industry, they lacked control of their actions. Humans had total control of the machines and provided all the direction, instruction, and information.

#### 2.2.2. Automation

Automation enables a reduction in human intervention in the production process. This is achieved through the predetermination of decision criteria, the development of sub-process relationships, and related actions and the embodiment of those pre-determinations in machines [62]. The integration of electronics, computers, control, and sensing elements into mechanized systems at the onset of the third Industrial revolution made it possible for machines to accept and execute instructions, giving them the ability to self-think, self-dictate, or more accurately, self-move. Automation is derived from two Greek words “autos” and “matos” meaning “self” and “thinking”, respectively. Notwithstanding the etymology of the word, automated systems have no initiative. They still operate within rule-based boundaries. They primarily execute pre-defined tasks assigned to them by humans—they just happen to execute them faster and more efficiently. The automation category is also dominated by industrial machines and processes that are managed by hierarchical, centralized, and rigid control systems. While they can be optimized for increased efficiency, they are very difficult to re-purpose and very poor at responding to change. The consequence is that they are unable to correctly respond when faced with unfamiliar situations, especially within dynamic environments, often necessitating the inevitable intervention of humans. Notwithstanding, the advantage of automation is that it reduces the demand for human (manual) intervention.

#### 2.2.3. Semi-Autonomy

Most advanced factories today are at the “semi-autonomy” level. Semi-autonomous systems feature programmable devices and machines. Unlike systems from the prior generation, they are smarter. They can better sense and understand their environments. Through the Internet of Things (IoT), they can interact with other entities, share data/information, and make some decisions within a volatile production environment leveraging machine/deep learning (ML/DL)-based technologies. The factors that helped facilitate this level of autonomy include the advent of the Internet, the maturation of computer networking technologies, the growth of the Internet of Things (IoT), and the rise of cyber-physical systems (CPS). Other factors include the cross-domain integration of modern operational technologies (OT) and info-tech/communication systems and the broader integration of advanced control devices (PLCs, PCs, PACs, etc.) into industrial systems.

#### 2.2.4. Autonomy

Autonomy is the highest level of system control by non-human actors within the industrial and manufacturing processes. It is the expected future state of the *Factory of the Future*, a state where very limited human intervention would be required for the full functioning of a factory [63,64,65,66]. Artificial intelligence (AI), which provides and represents computational procedures for automated sensing, reasoning, learning, and decision making, would be a critical part of autonomy in manufacturing [67]. At full maturation, the *Factory of the Future* would be managed by intelligent heterarchical (i.e., non-hierarchical, or unranked) control mechanisms, which would make it possible for the factory to respond well to volatility, unpredictable disturbances, or sudden change within a dynamic environment. While we are still years away from that day, there is the hope (think autonomous cars), that one day a fully autonomous factory will be technologically feasible. A factory at this level of autonomy would have the ability to execute work, predict failures, grow smarter, self-correct, and if necessary, recover or compensate for failures with little or no explicit intervention or instructions from a human. The attainment of this state would require the maturation of autonomic systems, which would involve the embedding of advanced cognitive and deep learning capabilities within multiple sub-systems inside the factory. The time frame for attaining this maturation level would likely coincide with the era of artificial super intelligence (ASI) or thereabout: the artificial intelligence level where the intelligence of a computer would have equaled the intelligence of a human in virtually all areas, including reasoning, planning, solving problems, thinking abstractly, comprehending complex ideas, learning quickly, and learning from experience [68].

Currently, human involvement in autonomic systems is significant. As hinted earlier, the goal of achieving manufacturing systems autonomy is still far off. Many more years of significant technological advancements would be required. Until then, the *Factory of the Future* will continue to exist within the wide chasm between automation and autonomy (i.e., Semi-autonomy). Full autonomy is a future goal that would require significant embedding of smart technologies within the manufacturing mainstream.

### 2.3. Industry 4.0 and the Future Factory

The term “Industry 4.0” (also spelled Industrie 4.0) has its origins in Germany [69]. It has been a subject of great intellectual and economic discussion within academia, industry, and government both locally and internationally [70]. It has also been a subject of multiple studies in the open literature [71,72]. Discussions around it first emerged in the literature sometime in November 2011 following a national strategic initiative by the German government aimed at the digitization, integration, transformation, and standardization of manufacturing. The US equivalent of this term is smart manufacturing [73,74]. Industrie 4.0 is a construct (idea or theory) of the Fourth Industrial Revolution that promotes the digitization of manufacturing. It seeks to improve upon the advances achieved in the Third Industrial Revolution, which included the adoption of computers and industrial automation. It is a step higher forward that includes the integration of interconnected systems of intuitive, self-regulating, smart, and autonomous entities that seamlessly exchange data, perform tasks, and work collaboratively. The goals of the upgrades are the improvement in productivity, flexibility, efficiency, and agility.

CPS [75] and IoT are the core technologies that make Industry 4.0 possible. Figure 2 shows the other enabling technologies that are contributing to its full realization. In the final report of the Industrie 4.0 working group entitled “Recommendations for implementing the strategic initiative INDUSTRIE 4.0” commissioned by the German Federal Ministry of Education and Research (BMBF) and aimed at securing the future of German manufacturing industry, Industrie 4.0 was defined as, “networks of manufacturing resources (manufacturing machinery, robots, conveyor and warehousing systems and production facilities) that are autonomous, capable of controlling themselves in response to different situations, self-configuring, knowledge-based, sensor-equipped and spatially dispersed and that also incorporate the relevant planning and management systems” [76]. Industry 4.0 encompasses a plurality of methodologies, technologies, and emergent trends [77]. At the very minimum, it is based on the technological concepts of cyber-physical systems (CPS) and Internet of Things (IoT) [78]. It is expected that Industry 4.0 will result in the exponential growth of data, improvement in operational effectiveness and a radical transformation of the landscape within several industry segments that will result in the development of new but nimble business models, build-out of highly customized products and the creation of very customer-centric services [76,79,80]. Interoperability, information transparency, technical assistance, and decentralized decision making were identified as the four pillars of Industrie 4.0 necessary for the development of the smart factories [81]. Industry 4.0 as a concept is evolving. It is embracing newer paradigms, technologies, and distinct arrangements and organizational structures for these assets. Beyond representing a technological transition from embedded systems to cyber-physical systems, it now refers to the networking, organization, and marshaling of intelligent objects (people, machines, and processes)—leveraging information, services, and communication technologies to help achieve machine and system autonomy, decentralized intelligence, and production optimization. Interoperability is about the ability of machines and other assets to seamlessly connect and communicate with one another (i.e., share data and information). Information transparency relates to the ability to access and acquire data and information and using same to build virtual models of the physical device or system. Technical assistance speaks to systems of non-human assets (robots, devices, and systems) working together to assist humans in solving problems. Additionally, decentralized decision making refers to the transfer of some resources and control to enable decision making at distributed locations within the network without a need for consent from a central control hub [81].

## 3. Research Methodology

The qualitative study of the *Factory of the Future* as a concept is a daunting prospect due to its multi-dimensional and cross-disciplinary scope. It must therefore be understood that a meaningful review of literature on the subject would require the interrogation of multiple research streams that are spread across a variety of domains and disciplines.

### 3.1. Research Approach

Given the aforementioned, reliance on traditional review methodologies for data collection, research trend tracking, knowledge syntheses, and objective viewpoint critique can be challenging, to say the least. For this reason, the semi-systematic (narrative) review approach was assumed to be a more realistic research approach. This approach caters to our need to implement a multi-disciplinary and multi-domain based reviews of literature in a less restrictive manner. This is in contrast to the traditional systematic review approach, which is primarily focused on the systematic appraisal, and synthesizing of primary sources around specific and narrowly scoped issues or topics. While a large portion of this review focuses on peer reviewed literature, it also draws upon findings from the gray literature (including industry reports, policy documents, white papers, etc.).

### 3.2. Search Goals

The authors’ primary search goal was to aggregate enough materials that could help the research team understand the *Factories of the Future* and the emerging technologies that are playing roles in making them smarter, faster, and more intuitive.

### 3.3. Data Sources

The primary data sources for this study included three digital sources, i.e., Google Scholar, Web of Science, and Science Direct. To acquire the relevant articles, these sources were searched using several keywords. Starting with relevant articles from our primary search, we were able to track down other important (and sometimes obscure) articles that did not show up in the preliminary searches using the *snowballing* search strategy. Snowballing requires a set of initial articles, called seeds, to start the process. There are two approaches to this methodology: *backward snowballing*, which selects papers referenced by the seeds, and *forward snowballing*, which selects papers that cite the seeds [82]. Snowballing allows researchers to complement their primary set of relevant papers with additional sources, often using the primary search results as the seeds or starting point. The additional papers that result from the snowballing search are often papers that do not satisfy the original search query or other relevant papers that are not present in the databases considered. Grey literature and relatively dated research often fall within this category [83].

### 3.4. Search Methodology

The first step was to search the data sources using certain keywords. Two main categories of keywords were generated. The first related to different manufacturing paradigms. This category included such keywords as “smart factory”, “intelligent factory”, “digital factory”, “industry 4.0”, “advanced manufacturing”, etc. They were used to aggregate content to enable an understanding of various aspects of the *Factories of the Future*, including the foundational principles, underlying concepts, and applications. The second category of key works related to the various emerging technologies that enable the *Factories of the Future*. Examples of keywords used in this category are “artificial intelligence”, “cloud computing”, “blockchain in manufacturing”, “big data in manufacturing”, “mixed reality”, “augmented reality”, and “virtual reality”. The search was focused on qualitative studies with abstracts or titles containing terms related to the different manufacturing paradigms and the key enabling technologies driving them. Once the relevant papers had been extracted, the titles and abstracts of the articles were read for the initial screening. In the third step, all the articles selected from the initial screening were thoroughly reviewed to ensure their relevance to the subject areas. To ensure that other relevant articles that did not satisfy the original search query or were not present in the databases used were not omitted, snowballing was used in the next step. Search results from the database query and snowballing search then made up the comprehensive list of papers from which the list of principles, characteristics, and applications was prepared.

## 4. Understanding the Future Factory

As the traditional factory becomes increasingly unable to address the manufacturing challenges of the 21st century, there is a need to to rethink the design of today’s manufacturing system. This would require the development of factories with capabilities which radically improve the predictability, reliability, efficiency, and security of manufacturing processes. This section explores the meaning and characteristics of a factory of this sort, otherwise referred to as the *Future Factory* or the *Factory of the Future*.

### 4.1. The Future Factory in the Context of the Manufacturing Ecosystem (ME-S)

Accenture *Global Strategy or Management Consultants)* defined the ***manufacturing ecosystem*** as, *“a network of industry players who work together to define, build, and execute market-creating customer and consumer solutions; defined by the depth and breadth of potential collaboration among a set of players. The power of an ecosystem is that no single player owns or operate all components of the solution, and the value the ecosystem generates is larger than the combined value each of the players could contribute individually.”* [84].

In the past, production literally happened “under one roof”. However, as products became more sophisticated, and users demanded more affordable yet higher quality products, it no longer makes sense to think of manufacturing as a one “factory-only” activity. Often, no single entity owns or operates all elements of a manufacturing solution. Now more than ever, every product that rolls off the conveyor relies on some form of third-party service or technology. It is the case that producing high quality and cost-efficient products and services is dependent on partnering with other providers and cooperating with different entities within the larger ecosystem, which incidentally is the key to efficiency. Due to the technological changes that have occurred in the past three decades, technology now has the power to reshape the production process by tightly connecting all relevant entities (humans, machines, and applications) to enable the seamless flow of the data required to provide the intelligence needed to drive efficiency. The goals of these networks are to exchange information, provide support, and explore meaningful ways to achieve shared objectives. These networks, loosely referred to manufacturing ecosystems (ME-S), often include partners, factory workers, suppliers, vendors, contractors, and even customers. Like a docking station, the manufacturing ecosystems (MES) is a hub on which the Future Factory sits, enabling it to connect to an interdependent group of interrelated entities and systems with which it routinely communicates and on which it depends for the supply of information and resources. As the Future Factory (FF) relies on the concepts of information and feedback, the seamless flow of information from and to entities within the manufacturing ecosystems (MES) is critical for its optimal performance. At an abstract level, each of the entities within a manufacturing ecosystems (MES) can be represented as an object which can be connected to one another and to various machines, devices, and sensors to help extract valuable information. Entities participating in a manufacturing ecosystem have the rare distinction of being able to access the decision making knowledge they need, when they need it, because of the power of technology. Entities in a manufacturing ecosystem collectively facilitate the transfer and eventual transformation of raw materials into finished products. It is difficult to fully understand the Future Factory (FF) in isolation from all other entities within the future manufacturing ecosystems (ME-S). In many manufacturing contexts, it is difficult to separate manufacturing operations, for example, from the supply chains that support them, setting the need to look at manufacturing through the prism of an ecosystem. Ultimately, the resilience of the smart factories and the third-party systems that support them will depend on reliable interactions between the two. Ultimately, the future factory would be the result of the total reorganization, connection, and efficient utilization of the means of production, including assets, processes, and people. It is less about the automation and digitization of individual elements or parts of the production process, and more about the connectivity or ties between a broad range of internal and external components and processes to enable real-time data sharing and information exchange, and intelligent feedback and responses between all nodes within the manufacturing ecosystem. The goal is to enable the achievement of smarter, quicker, and more efficient solutions. This would require seamless connection, efficient data sharing, and reliable transfer of information, in real-time, between different asset categories, including machine to machine (M2M), machine to device (M2D), human to device (H2D), machine to virtual twin (M2VT), factory to factory (F2F), factory to product (F2P), factory to human (F2H), and factory to supply chain (F2SC) within the manufacturing ecosystem. Each asset on the networked ecosystem is assigned an identity and connected to other assets in the network using Industrial Internet of Things (IoT) protocols and technologies.

### 4.2. Describing the Factory of the Future

The emergence of the *Factory of the Future* has been necessitated by sweeping globalization and unprecedented technological changes which have resulted in a very competitive and dynamic global marketplace. The resultant volatility has given rise to short product life-cycles, a “big ask” for on-demand production and increasing price pressures [85]. As a result, traditional manufacturing, with all its advanced enhancements, is fundamentally ill-equipped to meet the demand-pressures of the current environment. This state of affairs has necessitated a rethinking of current manufacturing paradigms. To benefit from the on-going technology advancements within the manufacturing space, multiple actors (the state, industry, and academia) are actively trying to shape the future of manufacturing, hence the various national strategies, paradigms (smart factory, intelligent factory, Factory of the Future, etc.), frameworks, and nomenclature proposals. An example of the modern manufacturing system is the Cyber Manufacturing System (CMS)-refer to Figure 3. Its backbone is the cyber-physical system (CPS) that enables data transparency, facilitates reconfigurability, and promotes process optimization.

Incremental changes to traditional manufacturing paradigms are fundamentally incapable of fully addressing modern manufacturing needs. As a result, manufacturing as a concept needs to be fundamentally re-imagined. As a craft, it needs to be radically re-engineered. Notwithstanding the obvious need and urgency to transform manufacturing as we know it, there is no consensus around a fitting nomenclature for a factory that can serve as a replacement for the traditional factory. In recent years, different manufacturing paradigms have emerged, including (but not limited to) manufacturing concepts such as the Reconfigurable Manufacturing System (RMS) [86,87,88,89,90,91,92], Flexible Manufacturing System (FMS) [93,94,95,96,97,98,99,100], distributed manufacturing [101,102,103,104], and cloud manufacturing [105,106,107,108,109,110]. In addition, terms such as smart factory [111,112,113,114,115,116], intelligent factory [77,113,117,118,119,120,121,121], and digital factory [122,123,124,125,126] represent emergent paradigms that seek to fulfill this need. However, none has seemed adequate enough to capture the approval of a plurality of stakeholders within the manufacturing community. This paper does not propose a unified nomenclature but instead uses a generic term (i.e., *Future Factory* or *Factory of the Future*) to help crystallize the general principles advocated by the most common advanced manufacturing paradigms. Additionally, note that *Factory of the Future* and *Future Factory* have been used interchangeably in this text and represent the future state of the factory during and beyond the Fourth Industrial Revolution (4IR). Given that the end goals articulated by many of these paradigms are similar, we will be drawing insights and inspiration from them to help characterize the Factory of the Future. The concept of the Future Factory upends the factors and elements of traditional manufacturing systems. The biggest challenge is being able to clearly define what the Future Factory is and to then transition those concepts effectively from theory to practice. For a start, the Future Factory is the nucleus of Industry 4.0 [127]. A proper articulation of the meaning of the Future Factory is heavily reliant on the contextual and comprehensive understanding of Industry 4.0 concepts, which is the reason why a Section 2.3 of this paper was dedicated to elaborating on Industry 4.0. The technical foundation on which it stands is the selective and strategic integration and exploitation of the unique advantages of a collection of emerging technologies, such as cyber-physical production systems (CPPS)/cyber manufacturing systems (CMS), the Industrial Internet of Things (IIoT), Internet of Systems (IoS), artificial intelligence (AI), smart robotics, cloud computing, cybersecurity, and big data analytics. Thus, the Factory of the Future, as a system of systems (SoS), should be designed to deliver key advantages that can potentially address the challenges confronting traditional manufacturing in the age of speed, volatility, and uncertainty [128]. While all the above-referenced technologies bring with them unique capabilities, the structural underpinning or technical framework that supports the Factory of the Future is the cyber-physical production system (CPPS). The future of manufacturing necessitates increased flexibility in product customization, process monitoring, product quality/process control, and service delivery. Hence, collaborative networks and cyber-physical production systems (CPPS) have been identified as the future of industry [129]. Administrated interactions and information flow among machines, people, organizations, and societies have been pursued as an ongoing research topic [130,131,132].

**Definition** **1.**
*Thus, a barebones definition of the Factory of the Future would be a dynamic and highly integrated network of cyber-physical production systems (CPPS) that communicate or interact with each other using the Industrial Internet of Things (IIoT) and the Internet of Services (IoS). From an enterprise perspective, the Future Factory is a vertically and horizontally integrated production system with an efficient connection between itself and the supply chain that supports it.*


**Definition** **2.**
*Aptly named, the Future Factory is not an “end state”, but a constantly evolving solution that continuously but strategically integrates new technologies and systems with the goal of creating a resilient, stable, and efficient system of systems (SoS) which can adapt to rapidly changing manufacturing requirements, fulfill dynamic customer demands (in a timely and cost-effective fashion), support customized mass production involving high variability (high product variety), provide options for small lot sizes, adapt to disturbances, and rapidly respond to change or otherwise recover from failure autonomously while eliminating the need for expensive post-process inspection.*


The Factory of The Future would be a key driver of manufacturing competitiveness across the globe because of the expectations of higher efficiency, lower production costs, mass customization, adaptability, and flexibility. Its true essence is the incremental improvements in how we design, manufacture, distribute, and service products.

The findings in this study must be seen in the light of some limitations. The first of these limitations is that it was performed using the semi-systematic or narrative review approach, which is less rigorous than the systematic review approach. The reason for the choice of the semi-systematic or narrative review approach was that the study involved the intersection of diverse disciplines and multiple technologies, a reality that hinders a full systematic review process [133]. The second limitation concerns the difficulty in standardizing all advanced manufacturing systems and factory types in terms of differences in paradigms, product types, production processes, automation levels, digitization levels, etc. The authors made great efforts to capture the high-level principles that underline a multiplicity of next-generation factory paradigms.

### 4.3. The Characteristics of the Future Factory

In its mature state, the Future Factory (FF) is envisioned as a production ecosystem that operates autonomously (i.e., requiring little or no human intervention) in various manufacturing operations, including production, logistics, prognostics, diagnostics, etc. From a technical point of view, all of these are made possible by a technological framework built using CPS, IoT, and intelligent decision support systems reliant on advanced analytics and knowledge learning methodologies [134]. Figure 4 graphically illustrates some of the most consequential characteristics of the *Factory of the Future*.

#### 4.3.1. Cognition

The massive generation, analysis, and fusion of data (big data analytics) into manufacturing operations means that agents or entities of the factory possess cognitive capabilities and can learn, plan, and interact, all while acting autonomously and in concert with other entities.

#### 4.3.2. Self-Control

The Future Factory would be able to respond to changing business demands and conditions in real-time [135]. It would have decision making and self-controlling abilities, leveraging the factory’s ability to extract and analysis large amounts of customer and machine data (cloud computing), and subsequently transmit useful information and actionable intelligence on a need-to-know basis to the various entities within the network. The ability of the production facility to have access to real-time information about changes to the business environments can help the factory to adjust operations accordingly and help reduce business uncertainties and meet customer demands. The ability of the factory to self-control would be possible because of its decentralized architecture that allows for the distributed storage and flow of information. Data and information would flow through (to and from) a decentralized hierarchy that includes multiples localized systems, hubs, machines, and related nodes within the network (including the products themselves).

#### 4.3.3. Self-Diagnosis (Machine Health)

They also self-diagnose and repair identified malfunctions without halting production or switching to a downtime mode.

#### 4.3.4. Connectedness

Inter (and intra) connection, automation, and networking of assets within and between various activity layers, including factories, the supply chain, and the community (IoT, blockchain). This characteristic is further discussed in Section 4.4.

#### 4.3.5. Agility

The factory of the future (smart factory) is flexible and has adaptable production processes [115].

#### 4.3.6. Context Awareness

The Future Factory is context aware. Context awareness in this instance refers to a system’s ability to self-sense, respond, adapt its behavior, and communicate based on information transmitted from its environment or gleaned from sensors embedded in several nodes or entities within the system. Sensors have become affordable and ubiquitous across factories and entire manufacturing ecosystems [136]. Key components of the system can negotiate with each other to both request and profile functions [137]. With technologies such as block-chain, radio frequency identification (RFID), and quick response code (QR code), the smart factory can systematically identify and track assets, products, and people, both spatially and temporally, making it possible for the factory to have real time knowledge of its current state.

#### 4.3.7. Modularity

Modularity is a property or quality of a manufacturing system that involves the use of different modules (components or sub-systems) as the basis of design or construction [138]. In a modular system, modules can be quickly composed (composability) or combined to form different configurations. Modularity is a need-based decentralization of the production system using sub-systems that can later be recomposed into different configurations [139]. Modularity reduces complexity. It helps break a system into different degrees of interdependence and independence [140]. In a modular system, tweaking of one or more components does not affect the functioning of all other components. Modularity enables such advantages as quick adjustments in production capacity and functionality (reconfigurable manufacturing systems (RMS)), higher throughput (dedicated manufacturing lines (DML)), and product variety using existing manufacturing systems (flexible manufacturing systems (FMS)) [141,142].

#### 4.3.8. Capacity for Mass Customization

Mass customization involves the efficient, cost-effective, and speedy customization of products and services at scale leveraging customer preferences (big data) [143,144,145]. This would be facilitated by real-time communication between products and the production lines. Traditional automatic identification and transmission technologies such as radio frequency identification (RFID) and quick response code (QR code) can play roles in enabling products and production lines to communicate in real-time. The information transmitted can be used, for example, to address bespoke customer requests (product customization) or to control the paths of products as they navigate through different manufacturing lines or stages.

### 4.4. Enabling Connectedness: Integration, Interoperability, and Consciousness

Of the eight characteristics of the *Factory of the Future* discussed in Section 4.3, connectedness is perhaps the most fundamental. For this reason, we will be expounding on the subject to help characterize the Future Factory. Unlike traditional factories, the Future Factory is very robust, fully connected, and agile. It can synchronously learn and adapt to changing conditions using information acquired from a constant flow of process and machine health data amassed from a variety of interconnected assets, processes, and systems. All of these are made possible because of the system’s “connectedness”. The connectedness of a factory is dependent on the degrees of integration, interoperability, and consciousness of the system. Several researchers [71,146,147,148] have suggested that these elements constitute key success factors for Industry 4.0 in general, and the Future Factory (FF), in particular. Integration speaks to the tight combination, amalgamation, or homogenization of the entire manufacturing network, and interoperability refers to the ability of these tightly integrated systems and components to communicate or talk to each other seamlessly and in real-time. Integration is central to interoperability. On the other hand, consciousness speaks to the ability to be aware (or cognizant) and responsive to one’s environment. Below, we will further discuss the three main functions elements of consciousness: integration, interoperability, and consciousness.

#### 4.4.1. Integration

Integration involves the tight linking of independent factories, processes, and product lifecycles into a core network that can communicate with one another, and share data as needed, while supporting distinct or shared technological and businesses objectives. The goal of integration is to enable structural cohesion, seamless flow of data, and the ability of independent entities to access actionable information (technical or enterprise-related) from an integrated network. The automation pyramid pictured in Figure 5 is a framework that shows the different level of automation in a factory or industry. It represents a strategy guide for the integration of technologies in a factory. The three different types of integration identified in the final report [75] of the Industrie 4.0 Working Group include (a) horizontal integration through value networks, (b) end-to-end digital integration of engineering across the entire value chain, and (c) vertical integration and networked manufacturing systems. By integrating all systems, the most up-to-date processes and product data are available at any time and can be shared with all entities (men, devices, and machines) on a need-to-know basis to help facilitate planning, production, maintenance, logistics, supply chain management, and customer service. Proper integration can result in a factory that is innovative, proactive, and agile. Such a factory would be able to adapt to market changes quickly, respond flexibly and rapidly to changing customer demands, and achieve/maintain competitive advantages over peers even in the most unpredictable business environment. One of the main features of the Future Factory is the data-integrated core capable of supporting truly automated value chains [105,149]. Currently, the degree of asset and system integration within most factories is very limited. Though some have achieved the full integration of the shop floor (field level), loopholes still exist from the integration layers up to the management level [150,151]. The goal of the Future factory is to connect all entities (machines, devices, people, systems, etc.) using standard communication protocols, and therefore, enable seamless interaction between all parties.

(a)*Vertical integration:* Vertical integration is the integration of all hierarchical physical and informational subsystems within a factory to achieve a flexible, self-managing (autonomous), self-organized, and reconfigurable manufacturing system that can respond to production uncertainties quickly, flexibly, and effectively.For example, if a client requests specific product customizations, the business development unit should not be on a call to engineering all day. All the information requested by the engineering department will already have been logged in the ERP system—essentially everyone has access to the same information (albeit different versions on a need-to-know basis). With this this level of transparency and the seamless flow of information, a vertically integrated system can be easily reconfigured to produce the customized product or manufacture small-lot sizes, at short notice without a significant technical or fiscal penalty. The integration typically cuts across different hierarchical stages beginning from the field (factory floor) and going right up to the enterprise resource planning (ERP) level. The joint implementation of all three integration options can result in a fully integrated factory comprising digitally connected entities (machines, people, products, and services). The overall goal is the integration of all digitized physical assets into an ecosystem that includes all elements of a local factory and all other entities or partners within their value chain. The ecosystems resulting from these integrations enable increased autonomy for system elements, decentralized control, improved efficiency, and transparency given the seamless transmission of data and information between all entities.(b)*Horizontal integration:* Horizontal integration involves the digital connection of a factory to other external entities and processes across its value chain. It is illustrated in Figure 6.This arrangement would include a digitally developed network of warehousing systems, transportation assets, and production facilities that feature ICT-based integration of everything from inbound logistics to production, marketing, outbound logistics, and services [134]. The connectedness makes it possible for real time data to be obtained, analyzed, and shared in real-time to facilitate rapid and accurate decision making. All nodes on the network can have access (on a need-to-know basis) to information about production status, inventory levels, available resources, and other critical information necessary for streamlined production. A factory so connected is said to be horizontally integrated. Horizontal integration can take place at different scales and at several levels. Optimal value can be created by a factory that can harness value from data gleaned from activities and processes, both internal and external to the factory. As part of the horizontal integration of a factory, suppliers, contractors, and even other factories, whether located in the same or different geographical locations, can be connected into an efficient ecosystem where there is seamless transmission of data and information. The efficiency gained by this arrangement can be mutually shared by all parties. For example, a factory can have regulated remote access to a resource (machine, device, software, etc.) within a company in another location. These outcomes can result in efficiency of scale, improvements in turn-around time, and higher productivity levels. Other advantages include transparency, better knowledge sharing, and improved communication.(c)*End-to-end digital integration of engineering across the entire value chain:* Though many factories have successfully digitized different aspects of their businesses, some have also ended up with segmented or siloed organizations where the systems of various units are unable to talk to one another. Though most manufacturing processes are supported by ICT, many of the systems and technologies that rely on them remain static and inflexible [75]. The result is that information flow is inhibited, throwing manual transmission of data across the individual aspects back into the discussion. Digitally connecting these different systems and technologies can be referred to as End-to-end engineering integration. More broadly, it involves the digital integration of all aspects of the value chain (sourcing, product development, production, logistics, operations, marketing/sales, after-sale services, etc.) to enable the seamless flow of data across the network for the purpose of delivering real-time information about production status to all stakeholders, enabling the development of new efficiency, supporting product customization [152], streamlining of processes, and a reduction in the unnecessary expenditure on manual activities. Figure 7 provides a graphical illustration of end-to-end digital integration.

#### 4.4.2. Interoperability

Interoperability is a very critical characteristic of the Future Factory, primarily because it involves the ability of different entities (machines, devices, applications, etc.) to exchange, process, and use data/information. Multiple definitions of interoperability exist [153,154,155,156,157,158,159] in the literature, but they all distill down to the ability of two or more entities to receive, process, and exchange content (data, information, or services) for their mutual interest, in a timely manner, without distortions or any form of semantic inhibition. The goal of the data, information, or service so exchanged is to help all parties to operate more effectively together [153]. Abe Zeid et al. (2019) [160] outlined two approaches to interoperability, i.e., syntactic and semantic: (a) *Syntactic interoperability* relates to data formats. The use of standardized data formats can support interoperability. (b) *Semantic interoperability is the human interpretation of content*. The commonly used standards for semantic interoperability are XML and the Resource Definition Framework (RDF). The authors also identified two types of interoperability: (a) *Factory interoperability (vertical integration)*: The ability of physical and informational subsystems within a factory to seamlessly communicate with each other. (b) *Cloud-manufacturing interoperability (horizontal integration)*: Cloud-manufacturing interoperability is defined as a form of horizontal interoperability where virtual elements of the production process can communicate with one another. This is broken down into three types: (1) *Transport interoperability* (related to the interoperability of data transfer/exchange using different protocols, i.e., Representational State Transfer (REST) over HyperText Transfer Protocol (HTTP), and Message Queuing Telemetry Transport (MQTT) [161]. (2) *Behavioral interoperability* refers to the system’s response when faced with multiple requests. (3) *Policy Interoperability* ensures cloud systems comply and conform to standards of stated regulations and policies. A huge cost penalty (upwards of $1Bn) was identified in the U.S. automotive industry due to the lack of interoperability across its supply network [162]. Interoperability is required to make complete integration possible.

Challenges associated with interoperability remain the main constraint to the full realization of Industry 4.0 within the manufacturing industry. For example, enterprise interoperability is major hurdle for many businesses that currently own expensive legacy equipment. Some of these organizations are reluctant to replace these equipment with newer assets because of the additional replacement costs, though they are open to getting the benefits of Industry 4.0 if it is possible to retain their existing assets [149]. Achieving interoperability in this case would require retrofitting these assets so that they can seamlessly communicate (or talk) to other assets (machines and devices). The ability of these assets to communicate with related assets within an industrial network is enterprise interoperability.

#### 4.4.3. Consciousness

Consciousness is the state of being aware of oneself and one’s environment. In the context of the Factory of the Future, consciousness speaks to the ability of a the factory and/or its respective elements (i.e., machines, robots, sensors, actuators, conveyors, products, etc.) to be able to connect and exchange information automatically with other components or systems within their ecosystem. It also includes the capabilities of predicting the behavior of connected systems, reacting appropriately, and optimizing the processes that take place within and around them. To make all of these feasible, conscious factories (including all related systems, machines, and products) need to be equipped with the capacity to monitor, detect, and control events. A factory that is conscious would be able to (for example) sense machine component degradation autonomously. It would also be able to predict (without prompting) a machine’s remaining useful life. Such a system would be self-aware and could self-predict, self-configure, self-maintain, and self-organize.

### 4.5. From the Automation Pyramid to a Decentralized and Distributed Network

The automation pyramid i.e., Figure 5 is a representation of the different layers or levels of automation in a factory. It helps graphically capture the integration of different technologies and the inter and intra-level communication pathways between them. A classical automation pyramid has five-level control layers that include the data, services, and functionalities which are hierarchical and relatively rigid.

Given the advent of Industry 4.0 and all the recent technological changes, the rigid, hierarchical model would not be adequate to represent the facts on the ground. Current manufacturing trends reflects a complete shift in paradigm from the previous era. Smart devices are ubiquitous within the manufacturing space, making embedded intelligence available at the extremities. The connectedness of assets is also unprecedented due to the advent of networking via open and global information networks such as the Industrial Internet of Things (IIoT) and the Internet making IT/OT convergence easier.

This level of networking was unavailable in previous automation technologies. With a variety of devices and machines connected in a mesh, laterally, horizontally, or vertically, it is difficult to tell in advance what device, process, or subsystem will interact with another and in what manner. More so, the existence of predictive analytics at the edges now makes responsive control possible without the need for clearance from monitors or assets domiciled in layers “higher up” in the automation pyramid. Cloud technology and virtual twinning also enable visualization of control. With such a preponderance technologies, easy adaptability of individual assets and an unprecedented increase in the overall intelligence of the system are easily realizable.

The classical automation pyramid is insufficient to represent these new realities, and it must be stressed that this hierarchical model is all but outdated. The attainment of the vision and goals of Industry 4.0 require more flexibility in the interconnectedness and communication between assets of different categories irrespective of the control layer to which they are currently assigned. It is for these reasons and more that there has been growing interest by scholars and practitioners in a gradual dissolution of the classical pyramid and the introduction of a more decentralized and distributed framework that would serve as an update or outright replacement of the classical automation pyramid [163]. Figure 8 shows one such model, laid side by side with the classical automation pyramid. The need for the realization of the full potential of Industry 4.0 requires these changes. Traditionally, IT-based enterprise systems (such as the ERP and CRM) and operational technologies (MES and SCADA) lived on different islands (i.e., different layers on the automation pyramid) with a firewall separating them.

While this model has worked well so far, it underutilized the potential value that integrating the two layers (and having them talk to each other) would have created. For example, data on customer complaints or preferences, maintenance data, and user behavior data typically captured with Customer relationship management (CRM) applications could potentially be useful in product redesign or improvement for product designers if they could have ready access to those through the enterprise resource planning (ERP) applications.

On the other hand, the ISA95 (IEC 62264) standards play an important role in the *Factories of the Future*. Apart from the network infrastructure that connects the office floor to the plant floor, information needs to be moved around the system to support manufacturing business and operations. ISA-95 is the framework that models this information. It defines the interface between control functions and other enterprise functions. It is technology-agnostic and is completely distinct from the network infrastructure that supports its flow. Table 1 shows the various standards aligned to the ISA-95.

## 5. Conceptual Frameworks and Reference Architectures

Levis A. (2009) [164] described a *reference architecture* as a layout, blueprint, or sketch that “provides current or future descriptions of a ‘domain’ composed of components, and their interconnections, actions, or activities those components perform, and the rules or constraints for those activities”. Another very practical description of a *reference architecture* is that proposed in the Systems Integration for Manufacturing Applications (SIMA) Background Study [165]. It reads as follows: “A *reference architecture* (a) pinpoints the functions required to accomplish a set of objectives in each domain, (b) identifies the types of systems [components] that perform, or support human agents in performing, the activities that implement those functions and (c) points out the nature and content of the interfaces required among those systems”. The point of the reference architecture is to identify the activities and the information flows required to accomplish required functions and to be able to standardize said activities to ensure their efficient and effective performance. Essentially, a *reference architecture* is a toolbox packed with recommended structures, relations, and integration designed to guide practitioners towards solution approaches that meet accepted industry best practices. Reference architectures typically minimize complexity, since they anticipate and addresses salient questions that would otherwise arise, thereby enabling practitioners to accelerate model development and deployment. They are usually defined at different levels of abstraction. They provide a common lingo or vocabulary that serves as the basis for shared communication during implementation, helping emphasize commonality amongst users. Typically, reference architectures also provide templates, reusable designs, and industry best practices that serve as scaffolds and building blocks (i.e., LEGO pieces) for new solutions. They also provide the interfaces (or APIs) and serve as frameworks for interacting with outside elements or functions that are related but outside the scope of the architecture. There are several reference architectures and standardization efforts that have been completed or are underway. Nation states, industry groups, and organizations have been at the forefront of these efforts. The multiplicity of these efforts has thrown up a patchwork of reference models with gaps and duplications. The challenge is that of developing a smart manufacturing reference model that maps to the other reference models and considers the peculiarities and interests of all major stakeholders. An ISO and IEC joint working group was created to develop a standard reference model of that sort. Starting in 2017, a joint working group (ISO-IEC JWG21) between ISO/TC184 (Automation systems and integration) and IEC TC65 (industrial-process measurement, control, and automation) commenced work to help develop a common reference model for smart manufacturing [166].

Some reference models include the Reference Architecture Model Industrie 4.0 (RAMI 4.0, Germany) [167], the Industrial Internet Reference Architecture (IIRA) [168], IBM Industry 4.0 Architecture (IBM) [169,170,171], Smart Manufacturing Ecosystem (SME) [172], Intelligent Manufacturing System Architecture (IMSA, China) [173], Industrial Value Chain Reference Architecture (IVRA, Japan) [174,175,176], Smart Manufacturing Standard Landscape (SM2) [166], Scandinavian Smart Industry Framework (SSIF) [166], KSTEP cube model (KSTEP, Korea) [166], NIST Smart Manufacturing Architecture [166,177,178] (NIST, USA), Scandinavian Smart Industry Framework (SSIF) [166], and ISO-IEC Smart Manufacturing Standard Landscape (SM2) [166]. As the industry begins to consolidate around certain reference architectures, smart manufacturing solutions developers, whose solutions have not been developed and implemented on industrially accepted reference architectures, might be faced with adoption challenges beginning in the very near future [179]. At the moment, there are several simultaneous standardization efforts, championed by different countries and organizations, that have been completed or are underway. The multiplicity of these efforts has thrown up a patchwork of reference models with gaps and duplication among them. The challenge is that of developing a smart manufacturing reference model that maps to the other reference models and considers the peculiarities and interests of all major stakeholders. An ISO and IEC Joint Working Group was created to develop a standard Reference Model of that sort. Starting in 2017, a joint working group (ISO-IEC JWG21) between ISO/TC184 (automation systems and integration) and IEC TC65 (industrial-process measurement, control, and automation) commenced work to help develop a common reference model for smart manufacturing [166]. At the moment, two of the most cited reference architectures are the *Industrial Internet Reference Architecture (IIRA)* developed by the Industrial Internet Consortium (IIC) and the *Reference Architectural Model for Industry 4.0 (RAMI 4.0) (IEC PAS 63088)* developed by the “Plattform Industrie 4.0” (Germany). Note that the Industrial Internet Consortium (IIC) is now the Industry IoT Consortium (as of August 2021). While these two architectures are very similar and interoperable, as can be seem in different studies in the open literature [180,181], we will be using RAMI 4.0 to illustrate the idea of a *reference architecture*. This reasons for this choice are its relatively high popularity and the fact that it was the basis for the development of several other important *reference architectures*. It has also been successfully mapped with nationally developed reference architectures, including those of USA (NIST), China (IMSA), Japan (IVRA), and even the Industrial Internet Consortium (IIC) (i.e., IIRA), and successfully introduced in most national and international standardization committees and co-operations.

### 5.1. The Reference Architectural Model Industrie 4.0 (RAMI 4.0)

RAMI 4.0 [182,183] is a reference designation system that describes Industry 4.0’s space using a cubic layer model. Quite simply, RAMI 4.0 is an architecture model of different elements of Industry 4.0 (i.e., information technology (IT), manufacturing, and product life cycle) integrated into a 3D layered model. The model makes it possible for the multiple elements and different internal connections within the Industry 4.0 ecosystem to be broken up as smaller subsystems and clearly represented in 3D [184,185].

Developed as part of the Platform Industrie 4.0 [167] working group’s standardization efforts for Industry 4.0, RAMI 4.0 is a three-dimensional (3D) map that integrates several elements and concepts of Industry 4.0 and how they relate to one another [186]. Different components of the model are abstracted and linked to established automation standards, such as IEC 62264, IEC 62890, and IEC 61512/ISA95. The model provides insight on how to approach the deployment of Industry 4.0 in a structured manner. It can be used to illustrate the different elements within the Factory of the Future, including classification, categorization, and logical groupings that provide insight on connections and potential data and information routes. As a communication tool, it provides stakeholders a framework for a common understanding and the exchange of ideas about the design and development of Industry 4.0-based systems. The RAMI 4.0 model cube shown in Figure 9 you provides three main axes for the dimensions of: (a) product life cycle and value stream (horizontal axis, left), (b) hierarchy levels (horizontal axis, right), and (c) interoperability layer (vertical axis).

#### 5.1.1. Horizontal Axis (Right)

The horizontal axis is orthogonal to the Life Cycle and Value Stream axis, and represents a hierarchy model of cyber-physical systems (CPS) management based on functional considerations on a layer-by-layer basis. It stems from the international standard(s) IEC 62264/IEC 61512. The hierarchical layers were adopted from the classical automation pyramid with enhancements that introduced the Product and Connected World categories at the beginning and end of the layer stack, respectively. Level 0 represents intelligent products. Level 1 and 2 are associated with the control and automation of the factory floor; Level 3 is related to the management of manufacturing operations; Level 4 corresponds to business planning and logistics; Level 5 is decision making systems at an enterprise level; Level 6 is related to connections to the cloud and interactions with (external) entities and associated stakeholders. The individual elements can be further described as follows:(a)**Level 0***(Products):* Refers to intelligent (communicating) products. They can interact with users and makers with or without embedded sensors, labels, or tags. Data pulled from these products enable product enhancement, maintenance, and future design improvements. Conversely, data can also be pushed to the products (example updates).(b)**Level 1***(Field Device):* The functional level comprising intelligent devices that enable smart and intelligent control of machines and systems. It includes sensors, actuators and all other devices required to protect, control, and monitor manufacturing systems and processes. Process and machine health data can be pulled from these assets. Actionable information can also be pushed to some (e.g., actuators).(c)**Level 2***(Control Device):* Represents industrial control systems that are responsible for the logical control of field devices. Examples include distributed control systems (DCS) and programmable devices, prominent among which are the programmable logic controllers (PLCs).(d)**Level 3***((Work Unit OR Station):* A lower-level element in the manufacturing architecture where production planning and scheduling (including supervision of machines) based on events and processes are performed. This is usually done using supervisory control tools such as supervisory control and data acquisition (SCADA).(e)**Level 4***(Work Centers):* Work centers are the highest-level manufacturing elements that perform and manage end-to-end manufacturing processes and functions, including planning, scheduling, and production activities. They typically include process cells, production units, production lines, and storage zones. Management execution systems (MES) and manufacturing operations management (MOM) applications are used to build traceable records of the manufacturing process, build supply chain visibility, and keep track of information of everything from labor to materials, machine health, product shipment, and job orders.(f)**Level 5***(Enterprise):* Strategic business decisions are made at this level. Enterprise resource planning (ERP) tools are commonly used. The enterprise is a collection of business functions operating together to set and implement and manage the realization of strategic business imperatives.(g)**Level 6***(Connected World:)* This level is one of two enhancements to the traditional automation pyramid. It is the level that enables connection to super-ordinate cloud services, the Internet of Things (IoT), and the Internet of Services (IoS), helping link assets in one organization to the assets in external organizations. The flow of data from the shop floor to plant operating systems (MES), business systems (enterprise resource planning, ERP), and then the external world (e.g., other smart factories or external elements of the value chain or supply chain).

#### 5.1.2. Horizontal Axis (Left)

This axis represents life cycle and value stream and is based on IEC 62890 (i.e., life-cycle management for systems and products) used in the industrial-process measurement, control, and automation domains. It is focused on product life cycle and value stream. The product life cycle captures the various stages a product undergoes, from its development to when it is decommissioned or removed from the market. Value stream encompasses all the actions or activities that culminate in the addition of value to a customer, from the initial request to value realization. This axis emphases the extraction, processing, and utilization of product life cycle information in addition to data capturing and utilization from all activities that culminate in value creation. This information be captured through digital representations of objects (products) using an administrative shell, an Industry 4.0 component. More on the administrative shell is in Section 7.1. The axis divides the product development and the usage process into a type and an instance phase. One of the design premises for type and an instance phase are the data type that can be captured during each phase. When an asset (e.g., product) is in the development phase (i.e., idea, research, design, development, testing, or analysis) it is in the type phase. As soon as it is production (physically produced) or service, it becomes an identifiable entity of that type and is transitioned to the instance phase. Once it transitions into production or service, it is in the instance phase. The point of the delineation of the phases is that different data types would need to be collected in different phases. For example, the necessary data required during the research and development of a product (say a car) would be different from those that need to be collected during its production or operation (i.e., while in use or in service).

An asset can be said to be in its type phase during its design (research and development).

Throughout the life of the asset, it is important that the relationship between the type and its instances is maintained (i.e., an instance of the asset might be required to mirror its type). Different data would need to be collected from the product at different phases in its existence. Software products are a particular example where updates on the type are often transmitted to the instance.

#### 5.1.3. Vertical Axis

The vertical axis represents the six interoperability layers [184,185]. The interoperability of the two horizontal components (left and right horizontal axes) can be considered in the context of these six interoperability layers. As an information and communication technology (ICT)-based representation system, it establishes a model for facilitating and implementing new features and choreographing the flow of data between different layers. The first three layers (Business, Functional, and Information) are related to functionality, and the lower three layers (Asset, Integration, and Communication) are associated with technical implementation. The interoperability layers are described as follows:(a)**Asset Layer:** The aggregation of all physical instances of assets and components required to provide functionality to the system. This would include physical objects, such as sensors, actuators, and devices. It would also include humans, products, plans, documents, applications etc.(b)**Integration Layer:** This layer manages the digital representations of physical assets and is responsible for the transitions from the physical to the digital world. It contains asset documentation, applications, and assets (i.e., HMI devices, QR-code readers, sensors, control systems, etc.) that manage the transitions, generate events from assets (e.g., equipment and machinery), and provide computer aided control of technical processes, system drivers, and other collaterals.(c)**Communication Layer:** Responsible for data integration and standardization of communication between the integration and the information layers. This layer includes standards, communication protocols, and services that support interoperability and integration.(d)**Information Layer:** This level manages and stores data in an organized fashion. It is associated with data services and standards that regulate the flow and exchange of information between components, services, and their functions. It also ensures consistency in the integration of different data formats and interoperability between components and services.(e)**Functional Layer:** This layer is responsible for production rules, decision making logic, and the provisioning and management of the run time and modeling environment for services that support business processes. It also hosts the descriptions of functions and supports remote access serving as a platform for the horizontal integration of various components and functions.(f)**Business Layer:** This level maps out the business model, links the various business processes, and hosts the business rules that the system must follow. Said rules are based on information drawn from the value stream, the supply chain, the regulatory regime, and subsisting laws. It also orchestrates (or arranges) services in the functional layer and receives events that help track the progress of business processes. Standard run-times for executable business processes are essentially provided in both the Functional and Business layers [187].

Notwithstanding its acceptance, RAMI 4.0 has not be widely implemented within industry. However, it has made a lot of in-roads within research and academia. To better illustrate the its utility, a few implementation examples and use cases have been included from the open literature [175,188,189,190,191,192,193].

## 6. Communication Standards and Technologies of the Future

*The Communication Layer and the ISO/OSI Layers:* The communication layer is only one of six layers in the vertical axis of the reference architectural model Industrie 4.0 (RAMI 4.0) [182,183]. Further treatment of this layer is being provided because of its relevance. It can be considered a crucial aspect of the factory because it provides the protocols and mechanisms necessary for the standardization of communication between different networked elements.

The goal of this layer is to arrive at a unified data format that ensures interoperability and to provide interfaces that can support data access: an outcome that would solve a protracted bottleneck that has stymied Industry 4.0 adoption. As the communication layer is heavily IT focused, it is complemented by the seven-layer International Organization of Standardization/Open System Interconnection (ISO/OSI) model. Figure 10 shows a graphical illustration of the seven OSI layers. The ISO/OSI model is a popular IT reference model that defines the seven levels in a complete communication system.

To enable multi-vendor interoperability and integration, it has adopted or recommended certain open communication technologies or standards for each layer (refer to Figure 10. Below we focus on only three that we think will have the most significant impact on the Future Factory. These include (a) *The Open Platform Communication Unified Architecture (OPC-UA)*, (b) *Time-Sensitive Networking (TSN)*, and (c) the *5th Generation Mobile Network (5G)*.

### 6.1. Open Platform Communication-Unified Architecture (OPC-UA)

Though not explicitly mandated, the Open Platform Communication Unified Architecture (OPC-UA [IEC 62541]) protocol [194] is RAMI 4.0’s recommended approach for implementing this layer due to its stability, scalability, and superior performance. The recommendation, beyond being very practical, has also received numerous endorsements [195,196,197,198,199] from stakeholders. As the standardized interface for communication between numerous data sources, the OPC-UA has become the de facto communication technology standard for many applications (both in industry and in academia) and is the focus of on-going research [200,201]. Some great characteristics of OPC-UA are that it is technology-independent, implements standard network protocols, and allows for easy integration into pre-existing IT networks. It also satisfies standard communication security requirements, supporting secure communication over VPN and across firewalls, through which it can establish seamless client-to-server connectivity.

### 6.2. Time-Sensitive Networking (TSN)

The OT systems used in many factories require specialized networks and protocols. Conversely, IT systems are typically general-purpose technology that rely on Ethernet networks. The unpredictable traffic patterns of the Ethernet-based “best effort” approach is unsuitable for handling time-critical data. The real-time transmission of critical data (e.g., process data) that is necessary for real-time control within the industrial domain requires latency guarantees that are not always available from Ethernet networks. The cycle time for transmission of time-critical data can be very minute, or sometimes as small as one second.

This disparity in standards and protocols often creates complications that significantly impact IT/OT integration efforts. There is also the problem of data volume and velocity. With the ubiquity of devices and sensors today, the amount of data running through some factories daily is enormous, sometimes overwhelming networks and storage systems. Though several real-time communication methods (such as Profinet IRT, EtherCAT, and SERCOS III), have been applied to address some of these problems, time-sensitive networking (TSN) is clearly a better alternative. Not only does it close the technical gaps between IT and OT, but it also addresses most other underlying challenges that have bedeviled the industry for a long time. TSN is an adaptation of the Ethernet IEEE 802.1 standard. It was designed to support time-sensitive networking and enables real-time, deterministic communication between networked assets. Figure 11, an adaption of the work of the Time-Sensitive Networking (TSN) IEEE 802.1 Task Group illustrates the four main characteristics of TSN and the common standards in each category. The four main components or characteristics of TSN include (1) resource management, (2) synchronization, (3) reliability, and (4) latency. These characteristics deliver several benefits, some of which include: (a) Support for time synchronization; i.e., all networked resources have a shared time reference. It is well-suited for the real-time control and the synchronization of high-performance machines, plus it offers solutions for efficiently managing network infrastructure with high bandwidth requirements. (b) Support for traffic scheduling (i.e., all networked resources observe the same rules for processing and forwarding packets within specifically reserved time slots). With this arrangement, different data traffic streams can be transmitted from a single standard open Ethernet network without any delays. (c) The ability to merge multiple industrial networks, including TCP/IP traffic, into one single physical wire. (d) The ability of mechanisms to temporarily interrupt the transmission of regular Ethernet-based traffic (i.e., based on the best-effort approach) to offer priority to critical data that are necessary gaining time-sensitive insights into processes and assets. (e) The ability to map across both the physical and data link layers, thereby reducing complexity and making implementation/management easier. (f) Compatibility with general purpose IT standards. TSN is a special type of Ethernet, and it is compatible with most IT systems. Its functionalities can be integrated into one standard open Ethernet network capable of supporting devices from different vendors, thereby ensuring interoperability and integration.

### 6.3. 5th Generation Mobile Network

The growing data traffic and bandwidth demands that are currently outstripping the ability of 4G (LTE) to cope because of the proliferation of smart devices, IoT devices and sensors, etc., is creating the need for ultra-low end-to-end latency in a variety of industries, including automotive and industrial automation industries [202]. The ability of 5G to deliver on these metrics is putting 5G in play within the industrial environment. 5G provides end-to-end, ultra-reliable, and ultra-low latency connections [203,204], which can help improve network efficiency in these settings. Several reviews [205,206,207,208,209,210,211,212,213,214,215] of 5G technology exist in the open literature. The convergence of 5G, AI, and IoT will result in major transformations in industrial control and factory monitoring (i.e., condition monitoring and failure prediction). With fast, efficient cloud-native 5G connections, factories can strategically redistribute computational power to allow for fast collection and processing of data with rapid AI inference at the edges. On the other hand, networked devices and applications can easily and reliably tap into edge resources without needing to access the core network. The analysis of industrial processes can also be performed with high degrees of precision, allowing for swift decision making whenever necessary. Given 5G’s promise of extremely low latency (no jitters) and high data rates for video transmission, there is an expectation of significant growth in innovation around the development of immersive and integrated media applications (e.g., mixed reality (MR), augmented reality (AR), and virtual reality (VR) applications which are becoming ubiquitous on the shop floor and beyond).

The 3rd Generation Partnership Project (3GPPTM) is a joint project that brings together several national standards development organizations (SDOs) with the goal of developing technical specifications for third generation (3G) mobile systems. The three main service categories for 5G New Radio (NR) as defined by the 3GPPTM are as follows: (a) Enhanced Mobile Broadband (eMBB), (b) Ultra-Reliable Low-Latency Communications (URLLC), and Massive Machine-Type Communications (mMTC).

(a)*Enhanced Mobile Broadband (eMBB)* services are geared towards applications that require high data rates across a wide coverage areas. Compared to 4G, they can handle large payloads and are stable over an extended time interval. Complimentary deployment of Enhanced Mobile Broadband (eMBB) alongside existing 4G broadband services could enable substantial improvements in traffic and the efficiency of the industrial network at the core network level.(b)*Ultra-Reliable Low-Latency Communications [URLLC]* is almost deterministic in time bounds on packet delivery. It is ideal for applications that require end-to-end security and where reliability and speed are critical, though bandwidth might not be as much. Mission-critical applications that require quick reaction times would fall into this category. As 5G URLLC delivers ultra-low latency and guarantees against triggering undesirable safety stops in production lines, it has been employed in automating factory processes and related power systems. For example, it has been used to run industry technical standards such as PROFINET. Industrial robots have become ubiquitous on manufacturing floors. The transmission of time-critical communication messages to them using Ultra-Reliable Low-Latency Communication (URLLC) might be necessary to accommodate for instances where decision time for responding to an incident or accident is almost non-existent. Combining 5G and MEC results in a significant reduction in network latency, which can improve the performance of previously tethered-only AR/VR, haptic, and tactile-based applications.(c)*Massive Machine-Type Communications (mMTC)* is a service that provides mainly wireless connections to massive numbers (tens of billions) of network-enabled devices that intermittently transmit payload sizes (small data packets) at low traffic [216]. While low transmission latency is not a requirement, it has low latency, is secure, is reliable, and is scalable. As mMTC transmit small payload sizes at low transmission rates and frequencies, they require lower energy consumption, making them well suited for battery powered, low maintenance end devices (i.e., low-cost sensors, smart meters, wearables, trackers, diverse monitoring devices, etc.). NB-IoT (narrowband IoT) and Cat-M1 (operated at 1.4 MHz bandwidth) are two 3GPP standardized technologies that support these network-enabled devices. NB-IoT supports ultra-low complexity devices with very narrow bandwidth and data rate peaks of approximately, around 200 kHz and 250 kbs per second, respectively. Conversely, Cat-M1 supports relatively more complex devices and operates at a bandwidth of 1.4 MHz, with lower latency and better location and asset tracking capabilities. Both can also sleep for extended periods and maintain excellent power-saving mode (PSM) abilities and extended discontinuous reception [217].

5G can also be integrated with other communication standards for improved effects. For example, to exact the optimal impact on industrial IoT services and wireless industrial networking, 5G is best integrated into the Ethernet-based industrial network with TSN. While dedicated to the control of data communication (synchronization and data stream prioritization), TSN can also help forward critical process data, ensuring that they arrive in time at different end points within the network. On the other hand, 5G can be dedicated to the transmission of non-real-time-capable data (i.e., monitoring, predictive maintenance, and energy optimization-type data, etc.).

## 7. Realizing the Promise of Industry 4.0 through the Digitization of Physical Assets

It is the case that there are often differences in device/equipment types (makes, models, age e.g., legacy equipment), communication protocols employed, and interactions between hardware, communication technologies, networking devices, and applications within factories. Due to these differences, making all entities (equipment, devices, applications, etc.) seamlessly talk to each other can be prohibitively difficult. Relying on a vast array of sensing devices, communication facilities, and services [218], these disparate systems generate and use large amounts of diverse and sometimes complex data types to perform various functions. Building systems that effectively store, manage, and analyze these data and ensure valuable use can be daunting. Two main hurdles need to be crossed to make this happen. First, all assets need to be physically connected. That feat has been achieved using technologies such as OPC-UA [219] (further discussed in Section 6.1 and related technologies such as Profibus/Profinet [220]. The second challenge is making all devices, irrespective of the “language” they speak, to both communicate and understand one another. As has been acknowledged in earlier sections of this paper, one of the key goals of the Industry 4.0 era is the digitization of industrial systems and processes with a view to harnessing intelligence from them to help improve operational efficiency, productivity, and value. However, the digitization of manufacturing systems requires the efficient connection of all assets within the production network and the seamless exchange of data between assets. It also requires the development of information models that can accurately describe all assets and information sources to enable semantic integration and interoperable exchange of data between all assets [221]. The development of robust data/information models is critical to realizing some of the more creative goals of the Factory of the Future, such as plug-and-play automation of production modules, easy reconfigurability of production systems to cater to small batch production of customized products, and self-organization of the production line. The Factory of the Future, as a highly dynamic environment, will feature fluctuations in the number and variety of nodes (assets). With the right technologies in place, adding, removing, rearranging, retrofitting, or upgrading a network of assets would not be a hassle [222].

The practical significance of AAS is large, as it can be used to transform a factory into an easily re-configurable manufacturing system (also known as plug and produce) [223,224] that is flexible [225] and is capable of dynamically orchestrating and allocating resources [226]. It can also have significant implications for preventive maintenance [227], product customization, and the design and development of upgradable production lines and order-controlled production. Implementing the AAS paradigm within a factory can make integration faster. It can also mean faster ramp-down and ramp-up of production lines, and maximization of production efficiency throughout the life cycle of a plant [228].

Though the promise of the digitization of production (Future Factory) is very compelling, there is still concern about how to implement these ideas in concrete terms using available technologies, techniques, and standards. It is instructive that whatever techniques and standards are adopted must be flexible enough to accommodate different device categories (age, variety, and types), application domains, and use cases, and must be able to transcend organizational boundaries [229]. RAMI 4.0 and Industry 4.0 components are two important and complementary constructs of Industry 4.0. They are both described in the Reference Architecture Model Industrie 4.0 (DIN SPEC 91345) [167]. Section 5.1 provides more details about RAMI 4.0. Industry 4.0 components, on the other hand, are made up of two main parts: (1) the assets and (2) the asset administration shell (AAS). This is further discussed in Section 7.1.

### 7.1. Industry 4.0 Components: Assets and the Asset Administration Shell (AAS)

I4.0 components (and especially asset administration shell (AAS)) is Industry 4.0’s recommendation for tackling the aforementioned implementation challenges. The entire idea of the I4.0 components is to encompass every asset within an administration shell.

#### 7.1.1. Key Elements of Industry 4.0 Components:

Some of the key concepts or elements of Industry 4.0 components are outlined below:(a)**Asset**: An asset is anything (physical or non-physical) within the production system that requires a connection to another asset or an Industry 4.0 solution, e.g., simple devices, components, machines, assembly lines, or even entire production systems. Other examples of assets include automation components, services, and even applications/software platforms. Each asset within the production system must be identifiable to the system (in the first instance), and to all other assets (including devices, systems, and services). To be considered compatible, each asset must have a set of defined properties and must be able to collect and share all relevant data to similarly networked entities (other assets; stakeholders, e.g., companies participating both in the value and supply chains) throughout its lifecycle. This means they each need to read, interpret, and understand all asset data, including identity (asset type, model number, etc.), operational status, and all other asset-related data.(b)**Asset Administration Shell (AAS):** Industrie 4.0 recommends the asset administration shell (AAS) as an important building block of the *Factory of the Future* [75,186,230,231]. Multiple articles in the literature provide reviews on asset administration shells (AAS) [232,233]. The asset administration shell (AAS) is a mechanism for digitally representing physical assets and other abstract entities. In practice, it helps provide descriptions of the properties and capabilities of an asset and serves as a platform for interaction between the asset and other assets.As an industrial application, a digital twin (DT) helps transform an asset to its digital equivalent, serving as a bridge between a tangible asset and the virtual or IoT world. A typical AAS holds identifying, operational, status, and technical information about the asset it represents, over its lifetime. It contains the communication methods and stores all asset related data [149]. Some of the information the AAS stores is related to the configuration of the asset, its maintenance record, or data related to its connectivity with other devices. Diagrammatically, an asset is enclosed within an asset administration shell (AAS), as shown in Figure 12.Each asset in the production system has its own administration shell, i.e., Figure 12. Two or more assets can be grouped into a unit [234]. The unit (much like an individual asset) can map to its own administration shell. A common administration shell i.e., Figure 13 can be used to manage the communication of multiple asset administration shells (AAS) at a higher hierarchical level. The configuration shown in Figure 14 is also suited for Factory-to-Factory communication. This architecture can allow the transparent and seamless flow of data between sister factories and other associated assets within a value or supply chain Figure 15 shows a different configuration of assets, each mapped to its own administration shell, and connected to other assets through open communication protocols to facilitate the seamless flow of dataBeyond acting as a store for important asset data, the AAS also serves as a reliable and consistent mechanism for managing data and related functions and services.

#### 7.1.2. The Anatomy of an Asset Administration Shell (AAS)

The Asset Administration Shell (AAS) is composed of a body and a header. Refer to Figure 16, a metamodel of a typical (3D Printing machine) Asset Administration Shell (AAS). The *header* contains identifying information that precisely describes the asset administration shell and the represented assets, plus related asset utilization information. High-level asset-related information stored in the header would include the asset descriptions, serial numbers, manufacturers’ identification, etc.

Other information could include information about the usage of the asset, its sub-components, and other high-level details about the administration shell. The *body*, on the other hand, contains information about the assets. It has two parts: a manifest and a component manager. The manifest serves as a directory that lists different sub-models. Sub-models (or partial models) are important features of the administration shell that represent different aspects of the asset they represent; each sub-model is standardized for each aspect of the asset, e.g., a description or capability of the asset. Each sub-model contains a structured quantity of hierarchically organized properties that refer to the asset’s data and functions (or capabilities). The properties have a standardized format based on IEC 61360. On the other hand, the component manager (or resource manager) administers the sub-models and helps link the information coming from the asset administration shell (AAS) to the larger asset network through the Industrial Internet of Things (IIoT).

### 7.2. Seamless Transfer of Data: OPC-UA, AAS, and Companion Specifications

Data exchange between entities in an industrial network is a critical feature of the Future Factory. The Open Platform Communications Unified Architecture (OPC UA) is an important technology for machine-to-machine communication. It defines data transport protocols and standardizes information modeling; however, OPC UA communication alone is not sufficient for seamless data exchange. Before OPC UA communication can effectively occur, the content of the data to be exchanged must be clearly defined. To make communication feasible and seamless, companion models need to be mapped onto OPC UA. OPC UA does not define data content but only serves as a framework for the description of the meta model. Though OPC UA defines a base information model, the actual definition of the data content for different domains is achieved using companion specifications or meta model, of which there are several (40+). Companion specifications make the definition of standardized exchanges possible within the framework of specific business functions. These domain-specific models make it easier to achieve interoperability between equipment and devices from different vendors. Companion information models follow standard syntax described in XSD file (XML Schema Definition), and therefore present data in a form that can be read by a computer program.

One such companion specification is AutomationML, which focuses on the engineering of automation systems. Thus, an implementation for a I4.0 components and its asset administration shell could potentially involve a technology combination involving the OPC UA (IEC 62541) and AutomationML (IEC 62714).

Within a typical manufacturing environment, there are different kinds of equipment from a diverse range of manufacturers, creating a situation where multiple communication protocols are in play within the asset pool. This creates communication problems for enterprises if the goal is IIOT integration (i.e., networking the assets and having them “talk” to one another), because until recently, different companies relied on different communication protocols and applications which were not interoperable. OPC-UA was created to solve this problem. and the adoption rate by industry has been impressive. It provides the additional benefits of secure communication (encryption and authentication) and a standardized interface. OPC-UA solves the operational technology (OT) communication conundrum.

The Future Factory integrates previously independent and discrete systems transforming them into a complex whole. It is literally the convergence of operational technology (OT) and information technology (IT). There must be a way to connect OT to IT. However, the IT and OT domains have significant differences, and there are unresolved integration and knowledge transfer challenges that need to be resolved. That is where the asset administration shell (AAS) comes in. It is the software/firmware component that transforms the physical assets into digital ones (or Industry 4.0 assets). That data content stored in the AAS is developed using the companion specification (i.e., AutomationML). The combination of OPC-UA and the AAS helps eliminate the discontinuity between the layers (OT and IT), enabling the seamless flow of data/information. Refer to Figure XYZ02 below.

### 7.3. Data Exchange: The Administration Shell and the Semantic Web

The AAS provides a consistent way of storing and managing all asset data, functions, and services so that they are readily available for manipulation, publication, and exchange between all network participants as required. Once connected, the AAS serves as a standardized and secure communication interface for sharing data and information about the asset’s identity, operations, and status with the production system’s network. Using a standard such as AutomationML, the AAS can be mapped to OPC UA, MQTT, or other formats [188]. Due to its standardized design, AAS can integrate the knowledge and semantics of multiple domains together, to help achieve component and cross-company interoperability across the entire value stream.

Though the current data exchange process for many manufacturing applications has greatly improved due to the use of predefined structures and keys, any real-world implementation is still dependent on laborious manual work. It requires a robust understanding of the AAS model, an appreciation of multiple terms/values, and time-consuming/laborious data mapping [235]. To alleviate the burden, some scholars [236] have recommended building a connection between current manufacturing-based data provisioning models (such as AAS) and the semantic web. Great reasons to consider this option is that sematic web representation formalisms such as RDF, RDF Schema, and OWL, are more matured, have more advanced data integration and formalization capabilities, and have the capacity to introduce logical reasoning to the asset administration shell (AAS). For these reasons, information models developed using these frameworks would be useful additions to information exchange systems [235] such as AAS. RDF and linked data principles have been successfully used to integrate different data types [237,238,239]. They are the basis for the development of semantic solutions or information models that have proved effective for seamlessly linking I4.0 components with generated data [240], hence helping improve the interoperability of production assets. To promote data exchange and enable semantic interoperability, RDF-based information models are aligned to important industry standards, such as RAMI [186] and IEC 62264 [241]. As a resource description framework (RDF), a standard model for data interchange on the Web makes the generation and transmission of data across networks easy. An additional benefit of RDF is that it makes data readily available on a standard interface using SPARQL3 (an RDF query language). A group of researchers [236] proposed adding a semantic layer to the administrative shell. As part of the proposal, the RDF is included as a middle layer that can be deployed to support interoperability between the data generated from both I4.0 components and legacy systems. The researchers envision the establishment of RDF as a common communication language (lingua franca) between assets within Industry 4.0.

## 8. Key Building Blocks, Technology Enablers, and Innovation Accelerators

Though many organizations now aspire to upgrade their manufacturing and business operations into full-scale *Factories of the Future*, knowing where to start or even what makes for an Industry 4.0-compliant factory is not always clear-cut. The fact remains that applications or instances of the *Future Factory* are process-specific, and therefore vary from one industry to another. However, there are certain elementary units or common building blocks that show up in one form or the other in most configurations of these factories. The number and variety of building blocks that constitute a specific *Future Factory* configuration will depend on the industry, and the unique process applications a company seeks to improve or optimize with its operations and processes. Understanding these key elements can prove helpful in properly characterizing *Future Factories* and smoothing the transition for industrial adopters. Figure 17 shows some of the most important core and periphery building blocks (or elements) of the *Future Factory*.

### 8.1. The Core Elements of the Factory of the Future

In the opinion of the authors, the trio cyber-physical systems (CPS), Industrial Internet of Things (IIoT) and digital twins (DT) constitutes the bare-bone elements of the current *Factory of the Future*. As will be shown later, several other technologies (including the cloud, artificial intelligence, AR/VR, blockchain, etc.) can be wrapped around these to extend the functionality, resilience, and integrity of the core elements or systems. Cyber-physical systems (CPS) and the Internet of Things (IoT) both enable end-to-end connectivity and support the transmission, transformation, and storage of data/information across different levels of the factory. The similarities and differences between CPS and IoT have been the subjects of many debates within the research community. NIST [242] performed an extensive review of these debates based on several references in the open literature [243,244,245,246,247,248]. On the other hand, digital twinning enables the virtualization of the system to enable data and information cloning and seamless transmission and manipulation. Based on our analysis, the three foundational elements of the factory of the future are: (i) cyber-physical systems (CPS), (ii) Industrial Internet of Things (IIoT), and (iii) digital twins (DT). These elements are further discussed below:

#### 8.1.1. Cyber-Physical Systems (CPS)

Baring the development of some more advanced system, the cyber-physical system (CPS) will be a central feature of the *Factories of the Future*. The *Factory of the Future* is essentially a network of cyber-physical systems (CPS). They bridge the cyber and physical worlds seamlessly. In CPS, computational and physical systems are intertwined, with the interaction between the duo being a convergence of computation, communication, and control. They also integrate sensing, control, networking, and computation into physical objects and related infrastructure [249]. The roles and uses to which they can be applied are virtually endless. Though things can get very complex, very quickly, multiple CPS can be combined into a super CPS or what can be referred to as a system of systems (SoS). CPS have been variously defined: One definition is that they integrate computation with physical processes, where embedded computers and networks monitor and control the physical processes [249]; and the behavior of the system is defined by both computational and physical components [250]. Rajkumar et al. referred to them as physical and engineered systems whose operations are monitored, coordinated, controlled, and integrated by a computing and communication core. They have also been referred to as hybrid systems that are simultaneously computational and physical [251]. Though there are different interpretations of what constitutes a cyber-physical system (CPS), there is nonetheless an agreement that they are the result of the integration of a computing nucleus and physical systems; and the computing core, like a central intelligence entity, orchestrates (monitors, coordinates, and controls) the operations of all elements’ nodes and entities within the physical or engineered system. In recent times, cyber-physical systems have permeated several elements of modern life.

There are multiple applications of CPS in a wide variety of industries. Some of these applications can be seen in self-driving cars, smart grids, robotic systems, unmanned aerials vehicles, advanced industrial control systems, and automatic pilot avionics [252]. In the case of the self-driving car, also known as the autonomous vehicle (AV) or driverless car, the computational core and the physical elements of the system are seamlessly incorporated to achieve the effective monitoring, coordination, and control of the vehicle. As a complex CPS, the vehicle combines a wide variety of physical nodes or sensors (GPS, radar, sonar, odometer, etc.) to perceive its environment. The sensory data collected from these physical assets are then analyzed and interpreted by advanced control systems to help identify suitable navigation paths and avoid collisions autonomously. Some autonomous control systems have even been able to make control decisions through knowledge acquired from steering patterns of human drivers acquired from video feeds from mounted-cameras and basic GPS-like maps. The seamless interaction of diverse components of the self-driving car to create value with little or no human intervention is a model for what is possible in the Future Factory (FF). Another relatable example is the smart phone, a mobile cyber-physical system which is a sub-class of cyber-physical systems. The smart phone is a composition of independently interacting physical components and a computing and communication core. The operations of the smart phone are monitored, coordinated, controlled, and integrated by the computing and communication core. The computational resources include a robust processing capacity with a local storage facility. Plus, there are the mobile operating system and smart phone applications. The physical components include several sensory input and output devices, such as cameras, GPS chips, touch screens, speakers, microphones, light sensors, and proximity sensors. The sensors gather distributed intelligence about the environment, including monitoring physical and cyber-indicators such as touch (touchscreen), sound (microphone), and the presence of nearby objects (proximity sensors). The computing elements communicate, gather, and analyze data from the sensors, developing actionable intelligence useful for carrying out more accurate actions and tasks or controlling or modifying the physical and cyber environments. A communication highway between the smart phone and other CPS, and network connectivity that links the smart phone and various servers and the cloud environment, are enabled by communication technologies such as WiFi, 4G, and EDGE. It is also important to note that there are structural similarities between CPS and the Internet of Things (IoT), such as the similar architecture; however, a major distinction between the duo is that there is a higher level of co-ordination between the computational and physical elements in CPS [253].

Smart manufacturing is a leading CPS application domain. As the backbone of most smart manufacturing solutions, CPS are the subject of extensive reviews and research work. For example, S.K Khaitan and J. D. McCalley [252] completed a literature survey on design techniques and applications of cyber-physical systems (CPS). The survey covers several important aspects, including the architecture and modeling of CPSs; simulation of CPSs; and tools and programming of frameworks for CPS and their verification, including model checking techniques to verify the correctness of their cyber–physical composition. *Cyber-Physical Systems: Foundations, Principles and Applications* [254] is a great resource for understanding the core elements needed to design and build complex cyber-physical systems. It also provides useful application examples. Examples of the industrial applications of CPS include the development of an automated warehouse system [255], the development of an industrial product service system [256], and the development of Cloud-Based Distributed Process Planning (Cloud-DPP), a CPS-based system that was designed to enable cloud-based distributed and adaptive process planning in a shared cyber workspace [257]. The system can generate machining process plans adaptively based on real-time information from machines. Another CPS-based solution worth further investigating is the FESTO pre-industrial system MiniProd, a multi-agent system running on EAS modules (modules with embedded control) designed for easy reconfiguration on-the-fly and self-configuration. It is part of an effort to study the issues of autonomy and adaptability at operational levels of the assembly system within the smart manufacturing domain [258,259,260,261,262]

#### 8.1.2. Industrial Internet of Things (IIoT)

Generally, a device is considered an object (or “Thing”) if it can use sensors and application programming interfaces (APIs) to connect, transmit, and exchange data over the Internet. Other acceptable characteristics of IoT devices include excellent power management, the ability to self-diagnose, and the capacity for configuration upgrades at low Internet bandwidths and within domains with poor network connectivity. Hence, the Internet of Things (IoT) is simply a growing network of billions of physical objects (or “Things”) that are connected to the Internet or to other devices that can be connected to the Internet themselves for the express purpose of data exchange or transmission. Almost every field of human endeavor can benefit in some way from IoT integration. Fields as diverse as agriculture, consumer electronics, and home appliance industries have already shown varying levels of adoption. Though the IoT evolved from the IIoT, the “Industrial Internet”, also known as “Machine to Machine (M2M)” or Industrial Internet of Things (IIoT), is a much narrower or limited version of the Internet of Things (IoT).

It has been successfully applied to manufacturing and other high stakes industries, such as aerospace, healthcare, defense, and energy. The “Industrial Internet” as a term was originally coined by General Electric (GE) in 2012. It refers to a system of connected, albeit uniquely identified devices, alongside intelligent analytics, that are able to transfer data over a network without requiring human-to-human or human-to-computer interaction. The connected devices are interrelated objects, including sensors, actuators, instruments, and other networked assets, such as computing devices and digital and mechanical machines. While IIoT supports sophisticated devices with advanced analytics and automation usually with high-risk impact, the high-volume general IoT uses simple applications and focuses on value creation in the low-risk impact consumer experience space. So important is the “Industrial Internet” that several organizations, including Bosch, DellEMC, General Electric (GE), Huawei, Microsoft, and Purdue University (College of Engineering), came together in 2014 to form the Industrial Internet Consortium to help accelerate the growth of the Industrial Internet.

#### 8.1.3. Digital Twins

Digital twin technology is a key enabler for the digital transformation of the traditional factory. One of the earliest references to “digital twins” as a concept dates to 2003, when one was first introduced as a virtual, digital equivalent to a physical product by Dr. Michael Grieves in the University of Michigan Executive Course on Product Lifecycle Management (PLM) [263]. Other such early mentions can also be attributed to Främling et al. (2003) [264] and Shafto et al. (2012) [265]. The digital twin has been variously defined as a sensor-enabled digital model [266] or a digital replica of a physical entity [267] that “mirrors the life of its corresponding [flying] twin” (Shafto et al., 2012) [265]; uses the best available physical models, sensor updates, fleet history, etc.; and can simulate the health condition of the physical twin, by continuously recording and tracking its condition during the utilization stage (Lee et al., 2013). The digital twin is also defined by its abilities to perform real-time optimization [268] and monitor and control its physical twin while being constantly updated itself, using data received from the physical twin [269]. The synchronous existence of a physical asset and its digital twin means that the boundaries between the physical and the virtual worlds are effectively blurred, ensuring that data are transmitted seamlessly between both entities [267]. Figure 18 is an example of an implementation of the digital twin of a robot arm. Though a relatively nascent technology, digital twin technology represents the next step in the development of intelligent products and is a key enabler in the digital transformation of traditional manufacturing. It makes it possible for physical assets to take on virtual identities and interact with other machines and people across diminished virtual and physical boundaries. A digital twin (also known as a “living” simulation) mirrors the current state of its corresponding physical asset and maintains its characteristics as the asset’s exact virtual representation by constantly learning, refreshing, and updating itself through inputs from human experts, machine-to-machine interaction, and continuous exchange of data with key elements of the physical asset, including sensors, actuators, and the like. The value of accurately capturing the current state of the asset is that critical outputs from the emergent model can be fed back to help optimize the performance of the asset and serve as a critical input into the simulation and prediction of the future state of the physical twin. Digital twins can be used to perform system optimization. They can also serve as sandboxes for testing new ideas or making informed production decisions. In such scenarios, they can be used as simulators where the possible outcomes of multiple production scenarios can be gamed, contemplated, or investigated before implementation to eliminate the cost of actual production testing or avoid the impact on or disruption of on-going production. They can also be used to evaluate the impact of modifying manufacturing parameters or using a combination of system parameters in a manufacturing scenario. Think of this as a typical of what-if analysis. The best performing options determined by the digital twin can easily be deployed to the physical twin through embedded PLCs and/or microprocessors for immediate implementation, saving time and cost.

The digital twin is also gradually maturing into a technology that could ultimately revolutionize structural health monitoring, anomaly detection, and the remote launching of maintenance services that allow products to self-heal. On an even broader scale, digital twins of different factories and those of their suppliers, contractors, etc., can be linked to establish virtual supply chain networks. The main benefit of the digital twin is its ability to integrate previously disparate models into an integrated set of interoperating sub-models that are able to communicate and transfer information with each other while simultaneously drawing on multiple data sources, including historical data, comparable data, and current (up-to-the-second data) to accurately mirror the state of the physical twin, determine failure trends, estimate failure timing, and correctly interpolate results that help predict its likely future states. The continuous monitoring of the system means that multiple sensors (including smart sensors) positioned around the physical twin can constantly feedback data that would contribute to maturing the models and improving their accuracy and reliability. One of the challenges of the Factory of the Future is that it will be comprised of hundreds (and in some cases, thousands) of machines and devices at the edge. Updating the firmware, configuration, or software in all these assets (machines and devices), within multiple platforms, would be very challenging without some form of automation. The digital twins of these physical assets (machines, devices, systems, or systems of systems (SoS)) [270] can serve as remote centralized hubs or connection boundaries for their wireless updates without physical human intervention. Another advantage of digital twins is that they can be used for the remote co-ordination and operation of machines, devices, and systems. Other possible uses are process monitoring and product tracking, not to mention that they can provide reliable alert and notification system functionality both for the manufacturing floor and for supervisory and managerial teams. They can also be used to support the provision of remote technical assistance, equipment maintenance, and repair and other technical support activities in combination with augmented and virtual reality technologies.

The digital twin as a concept and technology has a lot of promise in smart manufacturing. The main idea of the DT is the digitization and virtualization of physical assets to enable the simulation of their real-time states through modeling, simulation, and analysis. The capturing of the states thus enables a feedback loop that makes the prediction and control of their future states and behavior possible [269]. Designing subsystems within a smart manufacturing system can be relatively easy. However, designing a fully integrated smart manufacturing system, complete with all its subsystems, can be very complex. The complication arises from the difficulty in being able to predict the dynamics between all the complex couplings that together make up the multi-field physical system. To reduce cost and minimize error and cost, designers can flexibly understand system dynamics, run what-if analysis, and redesign smart manufacturing systems using digital twins. J. Leng et al. [271] proposed a framework to demonstrate how digital twin technologies are integrated into smart manufacturing system (SMS) designs to create digital-twin-based smart manufacturing system design (DT-SMSD). The framework referred to as the *Function–Structure–Behavior–Control–Intelligence–Performance (FSBCIP) framework* could potentially contribute to easing the complications in the process of concurrent SMS designing. Further research is needed for the development of unified models that effectively imitate every interaction and behavior of the all the manufacturing processes within the co-interacting subsystems in a smart manufacturing system. Open-source technology and open architecture have been recognized as potential solutions to SMS design [271]. Violeta D. and Wernher B, developed an implementation of the digital twin (DT) for a smart manufacturing application using an open-source approach. The Digital Twin Demonstrator (as it is called) gave rise to a high-level micro-services architecture and includes a few building blocks, including data management modules, models, and services. In their work, they also listed several commercial and open-source digital twin solutions that are worth mentioning. Some of these include the digital twin (DT) of a jet engine by General Electric (GE) that enables pre-procurement and pre-construction configuration of their wind turbines. This solution is based on the Predix platform [272]. Other DTL solutions include the PTC Windchill (smart PLM software), Build to Operate (aerospace and defense manufacturing management software) [266], and DXC (performance prediction solution for hybrid cars) [273]. Kamil Ž. et al. [274] discussed the creation of a digital twin for an experimental assembly system that was based on a belt conveyor system. The system also includes an automated line for quality checking. The paper provides a very good look at the DT implementation for a smart factory solution. However, the authors also reported data synchronization challenges (data transfer delays of up to one second) due to the cloud platform’s lack of support for storage of customized digital twins. Liwen Hu et al. [275] built a cloud-based digital twin (CBDT) using an information model and MTConnect protocol. The CBDT provides usage predictions and estimation utilities to improve efficient resource use for cyber-physical cloud manufacturing systems. In 2020, the Industrial Internet Consortium (IIC) released a white paper [276] that provided practical guidance on digital twins, including the definitions, benefits, architectures, and necessary building blocks for their implementation. The paper describes the technical aspects of digital twins, their design, standards and frameworks, and a high-level discussion of their applications in various domains. It also illustrates how digital twins are used to tackle the information silo problem within industry. First, the digital twin engine is used to centrally collect data from the disparate parts of the manufacturing system. That information is processed and then pushed to different parts of the system through integration interfaces, such as application programming interfaces (APIs).

### 8.2. Peripheral Elements of the Factory of the Future

Key enabling technologies (KETs) are emerging: high-tech technologies and solutions that have been permeating the traditional manufacturing industry, leading to industry-wide transformations [277]. Using case studies from Germany, Michael Rüßmann et al. [278] were able to identify the top nine technologies that constitute the building blocks of Industry 4.0. Several other authors have also identified similar technologies as major enablers driving the technological revolution. In this work, we will be focusing on some of these. The technologies and solutions that are the basis of our discussion include cyber-physical systems, Industrial Internet of Things (IIoT), cybersecurity, digital twins, cloud computing, artificial and cognitive intelligence, big data and analytics, blockchain, augmented reality, 3D printing (additive manufacturing), and autonomous robots. These technologies are weaved into the fabric of the factory and improve the intelligence of the system. Figure 19 shows some of these technologies illustrated as layers.

The Industrial Internet of Things (IIoT) will emerge as an important feature of the factory of the future. Currently, many Industrial facilities are increasingly reliant on the Industrial Internet of Things (IIoT). The connection and exchange of data among sensors, software, and other technologies that it make this possible help with the streamlining of operations, performance optimization, predictive maintenance, remote process monitoring, and online progress tracking. While the growth of the IIOT within the manufacturing industry will enable manufacturers to operate at unmatched performance and revenue levels, it will also create unprecedented susceptibility to cyberattacks in industrial systems and networks, the main reason being that connectedness can create vulnerabilities, since it opens more doors, exploitable touch points, and attack vectors or surfaces for not only cybercriminals but other bad actors, including individuals, groups, and nation states who might have ulterior motives or an axe to grind. The higher the number of devices and sensors connected through networking and Internet protocol (IP) addressing, the more the access gateways. Potential security breaches by malicious actors, or even insider threats, can portend grave technical and business risks if not adequately addressed. These vulnerabilities have been exploited in the IoT domain in the past. For example, hundreds of thousands of unsecured IoT devices were pulled into a botnet (codenamed Mirai) which aggregated their processing power to carry out large-scale cyberattacks that momentarily crippled major websites such as PayPal, Netflix, and Spotify. These cybersecurity challenges are even more complicated in the Industrial Internet of Things (IIoT) because both information technology (IT) and operational technology (OT) systems are being pulled into the Industrial Internet of Things (IIoT), even though operational technology (OP) systems were predominantly closed systems until recently. Furthermore, information technology systems have established cybersecurity protocols which do not work well with operational technology (OT) systems [279]. Note that information technology systems as defined in this paper would include computers, computer networks, electronics, semiconductors, and telecommunication systems; and operational technology (OT) systems include all software and hardware systems focused on the physical aspects of industrial production, including the monitoring and control of physical devices and machines. Operational technology (OT) systems are more likely to include outdated hardware components and legacy applications that have not been updated for years. There is also the challenge of very vulnerable communication protocols (i.e., Modbus and Profinet) that are used to control many sensors, controllers, and actuators. A March 2019 report by the Ponemon Institute showed that 90% of organizations dependent on operational technology (OT) experienced at least one major cyberattack within the previous two years [280]. Since it is unlikely that companies will immediately upgrade their decades-old equipment that is still functional, there is a need for more cybersecurity research and a paradigm shift to address most of these security challenges.

#### 8.2.1. Cloud Computing

The cloud has made it possible for data to be stored and accessed differently than was previously. The emergence of the cloud is one of the main factors driving the development of smart technologies. The cloud has forced a shift in the geography of data storage and computation. When “cloud” is combined with “computing”, it takes on an even more consequential meaning. Many scholars have attempted to provide some insight into the meaning and consequences of cloud computing to technological change. Armbrust et al. described cloud computing as, “both the applications delivered as services over the Internet and the hardware and system software in the data centers that provide those services”. Cloud computing has also been described as a unique computing paradigm that involves the provision of flexible, dynamically scalable, and often virtualized resources over the Internet [281]. As cloud computing is based on an on-demand service delivery model, it was referred to as a utility by a 2009 Berkeley Report, a reference that is not entirely surprising, given that cloud computing has been previously referred to as utility computing. Cloud computing is premised on the idea that IT services (computational power, storage, platform, and software) can be provided as utilities or services, much like electricity or water. It features ubiquitous, on-demand, and scalable access to resources. Under this arrangement, these resources are provisioned from a shared pool, obligating users to only pay for resources consumed (pay-per-use model). It eliminates the need for users to individually build and maintain complex infrastructure. The accessibility (location) to cloud infrastructure and how it is deployed or controlled (proprietorship) and by whom, vary from one cloud system to another. The four most common deployment models include public, private, community, and hybrid. The suitability of each model would depend on the specific needs of an organization. The cloud services are packaged in three main service models, viz., (a) *infrastructure as a service (IaaS)*, (b) *platform as a service (PaaS)*, and (c) *software as a service (SaaS)*. Quality of service (QoS) requirements between providers and users ensure that high-quality services are provided at competitive prices. Since the cloud is designed as a network of virtual services, deliverable over the Internet, organizations can access and deploy applications easily from any location. End-users (or last-mile consumers) can also seamlessly access information or personal data remotely. Due to its almost unlimited digital storage capacity, the cloud will be critical for the monitoring, tracking, management, and storage of an almost endless stream of data flowing from different nodes within the Industrial Internet of Things (IIoT). Pairing the cloud and the Industrial Internet of Things (IIOT) with a Future Factory enables the full integration of all key elements of the digital manufacturing ecosystem, which can then result in data and information sharing that should ultimately lead to gaining better understanding of how to improve productivity and ramp-up efficiency.

In the context of manufacturing, data acquisition or capture primarily occurs at the nodal level. It typically involves the autonomous capture of data from production equipment, machines, devices, or systems using embedded or connected sensors and related hardware. Large amounts and often expensive network bandwidth are required to transfer massive amounts of data captured from devices to the central cloud for deep learning (DL) model training and inference. Some results of inadequate network capacity when confronted with massive amount of transferable data are low throughput, delayed transmission (i.e., high latency), and poor network performance. Latency, which is the round-trip time required for data to be transferred to and from the cloud, needs to be as low as possible for systems to function optimally within the manufacturing network.

The long-established trend of capturing and transferring data from the factory floor to the cloud is increasingly becoming untenable due to the high latency and low bandwidth issues associated with the massive amounts of data captured daily by the ever-growing number of IoT devices now available within manufacturing ecosystems. Furthermore, many time-sensitive operations within the factory have strict delay requirements (in some cases, a few milliseconds) [282] that cannot be met by reliance on the centralized cloud. Against this backdrop, it is instructive to look at the three levels at which storage and computing can occur:(a)*“Cloud-Only” Computing:* Cloud services make it possible for businesses to increase storage and computing capacities on-demand and on the fly without the need to invest in new infrastructure, applications, or IT personnel. Eliminating concerns about the availability of IT resources enables companies to focus on innovation and creating business value while simultaneously cutting down on maintenance and administrative costs associated with managing their own IT infrastructure.(b)*Fog Computing:* Another compute paradigm that has since emerged is fog computing. Hierarchically, it stands mid-way between the cloud and the edge and lives on the LAN. Programmable fog nodes serve as traffic hubs where decisions about the routing of data, inter-node peer-to-peer (P2P) communication, and service orchestration are made. They facilitate the decentralization of control and facilitate increase reliability, efficiency, and flexibility [283]. While fog computing and edge computing have been treated in some texts as interchangeable terms or overlapping terminology, it is important to clarify that they are indeed interrelated but also different in many ways. The OpenFog Consortium Architecture Working Group (now part of the Industrial Internet Consortium), an academia–industry group dedicated to the acceleration of the growth of the industrial Internet, called attention to some of the differences in a report entitled “OpenFog Reference Architecture for Fog Computing” [284]. While both computing paradigms bring processing power and intelligence closer to the data source, the major difference between them comes down to where data processing is performed. In the case of edge computing, data is processed directly on the devices on which the sensors are embedded or attached, or on some gateway device within their proximity. In the case of fog computing, processing is performed by processors connected to the LAN (i.e., a micro-data center) or within a LAN hardware. In either fog configuration, processing occurs further away from the sensors and actuators than in edge computing. Further to this, most edge devices only process data collected at one touch point, whereas fog computing is about processing data aggregated from multiple devices. Thus, the fundamental construct of the fog architecture is the aggregation and high-level processing of data and the almost instantaneous return transmission of the acquired intelligence.(c)*Edge Computing:* Edge computing has since emerged as a viable alternative to the “cloud-only” or central cloud computing standard or architecture [285]. This distributed computing paradigm involves the transfer of computer power, networking, application services, and data storage capabilities to where they are most needed, which is at multiple decision points that are usually as close as possible to the data sources [286,287]. It also enables dynamic monitoring and control of manufacturing processes [288]. By building in flexibility around where computations can be performed and extending cloud computing standards to the far reaches of the edges of the network, many of the latency, bandwidth, and data throughput issues that have bedeviled cloud computing can be addressed [289]. The proliferation of edge computing as a concept has been facilitated by the growing adoption of faster networking technologies, such as 5G wireless; the integration of edge devices into manufacturing IT and OT networks; and the connecting of all these using IIoT. In this arrangement, algorithms can now run locally on edge servers or gateways, and data can be processed at a high level; then, some forms of analytics can be reported so that insights are provided in real time and human and machine queries are responded to in seconds. Some of this intelligence can be used to actuate other connected devices or systems where necessary, and actionable results can be instantly made available to workers on the factory floor and executives in offices. This is reminiscent of the autonomous vehicle, whose systems require instant feedback to make travel decisions, even while the vehicle is in motion, and in some cases even at high speed. Edge devices (nodes) enable edge computing by providing entry points into manufacturing core networks. They are usually mobile or fixed assets, often embedded or connected to machines or equipment. They are typically distributed throughout the factory floor and other remote locations, such as nodes across a wide network or stars strewn across a dark sky. Large IIoT operations, such as those in manufacturing facilities, typically host hundreds (or even thousands) of edge devices (nodes), which together form a network of edge devices that recognize and communicate with each other. The edge devices continuously and autonomously collect, process, and broadcast data, which provides significant visibility and awareness about events across the network. Some edge devices serve a dual purpose as sensing devices for capturing sensory information and actuators that can trigger or control other devices or systems. Some common examples of sensors and actuators in edge devices within manufacturing facilities include: (a) Sensors: pressure and temperature sensors; real time location systems (RTLS); cameras; near-field communication (NFC) sensors; light, proximity, motion, acoustics, and radio-frequency identification (RFID) sensors; ultrasonic sensors; flow meters; and fluid sensors. (b) Actuators: hydraulic and pneumatic ones, switches, relays, programmable logic controllers (PLCs), motors, and light and acoustics actuators. Under the edge computing arrangement, data are processed, and analysis results are distributed by the same device used to acquire it or by a nearby server instead of a centralized cloud. The results of implementing edge architecture include the ability to process and store data faster, improved application performance, low latency, and significant reductions in bandwidth cost. Notwithstanding, it is important to note that edge computing does not eliminate the need for deeper data analytics, large data storage facilities, and extended archival capabilities, all functions that the cloud is better suited for. The main advantage of edge computing is its capacity to reduce the compute requirements and data volume that must be transferred to data centers or cloud-based locations within short notice. In the *Future Factory*, it is expected that more complex data processing will be performed at the edge, as new system modules that incorporate advanced artificial intelligence functionalities are built into them. Edge computing has also helped in the management of many security and privacy related concerns within industry.

Pending the development of better technologies and more advanced architectures, the cloud computing/data analytics needs of the *Future Factory* can be met using a hybrid architecture that relies on or one or more of the computing paradigms discussed above. Examples of such hybrid architectures include:(i)*“The Cloud-Only” model:* In the cloud-only model, no intermediate processing of data occurs. All data captured by multiple sensors are transmitted to the cloud, where 100% of the processing occurs, before the results are pushed down to all the sources that require the intelligence.(ii)*The Cloud–Fog Computing model:* In this model, data from multiple sensors and devices are transmitted to the fog gateways. Depending on the urgency of the request, some high-level processing of data occurs at the fog layer, and intelligence pushed back to the various nodes (machines and humans) in real-time. Non-time-sensitive data and some pre-processed data that require further (deep learning) processing are transmitted to the cloud.(iii)*Cloud–Fog–Edge Computing Model:* This is a massive, distributed computing infrastructure that consists of three inter-connected computing tiers (cloud, fog, and edge).

All data acquisition occurs at the logical extremes of the network using edge devices. Some instant, high-level processing of data occurs at this tier (the edge) to provide time-sensitive, real-time responses from entities (man and machine) at different nodes. In this configuration, the fog layer not only serves as a distribution hub for resources and services between the edge and the cloud, but also stores and performs high-level data analysis of data from multiple sensors at different edge locations while providing a low-latency network connection for the transmission of data and responses back and forth between the edge and the cloud. Unlike the edge, the fog layer is best suited for analytic operations that require real-time analysis of data from multiple data sources (e.g., several edge devices). The cloud tier is where the most robust deep learning processing operations occur. It is the tier where all non-time sensitive and pre-processed data arrive for thorough, deep, and final analysis. The cloud also has massive and scalable data storage and archival capabilities. Figure 20 aptly captures the Cloud-Fog-Edge layer architecture including related infrastructure and services.

#### 8.2.2. Big Data Analytics

Data are the fuel that drives digitally transformed factories and are fast becoming the most consequential assets in manufacturing as the factory of the future begins to take shape. As part of this digital transformation process, some manufacturing organizations have been able to successfully connect their numerous fully automated manufacturing facilities (alongside all their production equipment) located in different global sites into a central cloud, resulting in manufacturing architectures that are IoT-native, fully digital capable, and broadly cloud-based. These architectures, which can qualify as benchmarks for Industry 4.0, enable seamless data sharing, collection, and exchange across enterprise resource planning, manufacturing operations management, and production life-cycle management processes, thereby enabling coherent feedback systems that leverage data analysis outputs for the optimization of manufacturing operations. These factories will plausibly grow ever more intelligent due to the exponential amounts of sensed data that will flow into servers and data reservoirs because of the continued digitization of industrial assets. However, no actual benefit would accrue from the possession of these vast amounts of data if they were not properly analyzed and the accruing intelligence distributed to all necessary end-users and connected systems in formats that make sense to help improve processes, productivity, and competitive advantages. Ultimately, deriving critical intelligence that can help make correct inferences and consequently acquire optimal value from data would require advanced analytics and the ability to effectively present results in formats that are easily decipherable by appropriate systems or visualization formats that are meaningful and can be effortlessly comprehended by end-users, both executives in the offices and technical personnel (engineers, technicians, fitters, etc.) on the shop and manufacturing floor. Terms such as data warehouse, data lake, edge, modeling, and optimization, are all examples of words associated with data analytics.

#### 8.2.3. Artificial and Cognitive Intelligence

The Future Factory is a highly dynamic system that is comprised of several interconnected and sometimes co-dependent sub-systems that are subject to a wide variety of nonlinear and stochastic activities [290,291]. These assets also generate huge amounts of data which potentially contain useful operational and strategic business insights. Unfortunately, only a fraction of these data are currently analyzed by various traditional factories, due to operational and technical constraints. With the increasing complexity of today’s factory, many traditional methods often used to address a lot of common production issues (such as process variability, root cause analysis, early detection of quality defects, degradation monitoring, and process control) are becoming increasingly inadequate. Due to its many successes in a variety of industries, artificial intelligence is increasingly looking like a credible alternative for addressing a variety of manufacturing challenges because of its robust portfolio of solutions and its incredible ability to process vast amounts of manufacturing data, making it possible for companies to transition from reactive to highly accurate proactive (and even predictive) decision making. Several research-based concepts, mock-ups, test-bed prototypes, and even factory-ready artificial intelligence solutions have been developed or built in recent years.

*However, what exactly is artificial intelligence (AI)?* Artificial intelligence (AI) is as an inter-disciplinary discipline [292], a set of practices and a variety of systems or tools that model and/or exhibit intelligent behaviors, such as perception, reasoning, decision making, the ability to predict, and even the ability understand context. The mimicry of human cognitive functions is at the core of artificial intelligence (AI). Reason, interaction, and learning are the three key attributes of a typical artificial intelligence system. Due to the implied similarities in intellect, comprehension, and abilities, artificial intelligence (AI) is said to possess a certain kind of machine intelligence [293] that could equal and possibly outmatch human intelligence [294] in certain respects. AI can either replace or augment human abilities. While there are several sub-fields within artificial intelligence (AI), (e.g., machine learning, deep learning, natural language processing, computer vision, expert systems, cognitive computing, etc.) it is not easy to make clear distinctions between them because of clear overlaps in their relationships. To better understand AI, we focus on the two main sub-fields, i.e., *machine learning (ML)* and *deep learning (DL)*, that drive performance in all the other sub-fields.

ML has been dubbed the workhorse of artificial intelligence (AI), and its applications are ubiquitous across multiple industries. It has been particularly useful as an effective tool for evidence-based decision making [295]. Traditional ML involves a process of training a system by exposing it to examples of desired input–output behavior instead of explicitly programming it. ML is used to build assets or systems that can automatically improve through experience. The way it does this is by learning from data. Learning from data means being able to extract information accurately and quickly from raw data and to be able to make reasonable inferences. To do this machine learning (ML) relies on different purposes or task-specific algorithms. Figure 21 shows different types of traditional ML algorithms. There are no one-size-fits-all algorithms that can solve all ML problems. ML algorithms exploit meaningful relationships within datasets to solve complex production problems. In his 1959 paper [296], Arthur Lee Samuel, an American pioneer in the fields of computer gaming and artificial intelligence, noted that, “A computer can be programmed so that it will learn to play a better game of checkers than can be played by the person who wrote the program”. This statement underscores the power of machine learning (ML), i.e., its ability to learn what to, to do it automatically once the lesson is learned, and to in fact, improve its performance and accuracy over time. Applied ML solutions do not only learn from data but become more accurate and useful over time by leveraging knowledge acquired from new data in the course of the use of the solution [297]. Vast amounts of data are generated in manufacturing. A great advantage of machine learning is that it can analyze large amounts of complex manufacturing data and quickly make meaning of it. There are numerous reviews of machine learning (ML) and deep learning (DL) techniques/applications in manufacturing [298,299,300]. The emergence of low-cost computation, next-generation computing architecture (particularly graphics processing units, GPUs), the availability of data, and the development of sophisticated algorithms are at the root of the rapid advancements in machine learning (ML) [301].

***Deep learning*** is a special type of machine learning (ML) that is based on artificial neural networks (ANN) [302]. The adjective “deep” in “deep learning” refers to the multiple network layers that are common in DL. DL algorithms use neural networks with multiple processing layers (often more than three layers) [303] that learn data representations at multiple abstraction levels [304] by optimizing some unsupervised criteria [305]. After the input layer, every subsequent layer within the network produces a distinct representation of the observed patterns based on inputs received from the previous layer. Ultimately, the algorithm achieves its results by progressively extracting high-level features from one representational layer to another. This intuitive stepwise feature extraction process results in slightly more abstract representation of the input data, the deeper into the neural network the data flows. Various deep learning (DL) approaches have been reviewed in the open literature [302,305,306,307,308]. Deep learning (DL) is very scalable, and performance improves markedly as more data become available. Some common deep learning (DL) application areas include computer vision [309,310], natural language processing (NLP) [311,312], speech recognition, machine translation, etc. Of these, computer vision is perhaps one of the areas where DL has had the most impact. Computer vision application areas, such as image classification, object detection, action recognition [313,314], motion/visual tracking [315,316], semantic segmentation [317,318], and human pose estimation [319,320], are now common on many factory floors, in the value chain, and in supply chains of many industries. The increase in popularity of DL-based solutions is in part because of the astounding human-level results they have delivered [304]. One of the major differences between traditional machine learning (ML) and deep learning (DL) is in how representations are learned from the raw data. Unlike traditional ML, DL can perform automatic feature extraction (i.e., feature learning) intuitively. While important features are manually extracted in traditional machine learning (ML), deep learning (DL) achieves relatively higher accuracy classifications using general-purpose learning procedures that rely on automatic extraction of high-level, non-linear features from raw data, all with little or no human intervention. This is particularly helpful considering that 80–90% of available data today are unstructured in nature.

***Applications of Artificial Intelligence (AI) in Manufacturing:*** The typical manufacturing system integrates several elements, including machinery (including machines, robots, conveyors, tools, fixtures, and related hardware), material handling systems, information handling systems (computer systems), and human workers. All these systems, the technologies that drive them, the processes they support, and the strands that connect them can all potentially be infused with AI solutions to increase efficiency and ensure the “optimal control” of material flow, efficient use of energy, and ultimately, the cost-effective creation of value. Amongst many benefits, artificial intelligence (AI) eliminates or replaces time-consuming and sometimes risky traditional practices, facilitates access to data, and enables effective execution of manufacturing tasks.

Artificial intelligence (AI) has domain-independent characteristics and has permeated many industries. AI-based solutions have been implemented for all manufacturing processes (design, production, maintenance, assembly, etc.) and the associated supply chain [321,322]. It has the potential for game-changing impacts on manufacturing in the long term.

AI technologies provide opportunities to maximize the value locked up in the vast troves of data generated daily within factories [323]. It facilitates autonomous and intelligent analysis of real-time and historical data, enabling smart and informed decision making [324]. This allows the factory and its sub-systems to respond in real-time to changing demands and dynamic conditions streaming through the PLM systems [325,326].

The two aspects of manufacturing that have experienced the most infusion of artificial intelligence (AI) solutions are machinery maintenance and quality. The most focus is directed towards advancements in overall equipment efficiency (OEE), growth in production yield, increases in uptime, and improvements in quality and consistency.

AI-based solutions enable automatic evaluation, monitoring, and real-time insight into equipment condition to help minimize unplanned equipment downtime and expensive maintenance costs. They are also able to forecast when operational equipment is likely to fail to help guide maintenance scheduling. Artificial intelligence (AI) also provides visibility across manufacturing cells, lines, and the supply chain.

It has been used for inspections [327,328,329], diagnosis [330], anomaly detection, and predictive maintenance [331,332]. To improve product quality, artificial intelligence (AI) solutions have been used to automate defect detection processes by automatically verifying product quality and providing insight into quality issues, hence reducing waste and enabling production improvements. Many ML-based fault/defect detection [333,334,335,336,337,338] and quality monitoring approaches [336] have been proposed in the literature. Several examples of fault diagnostics [335,339,340,341,342,343,344,345,346,347,348,349] are also available in the literature. Bayesian approaches that enable root cause analysis of quality issues [350,351] have also been proposed. It has also been used for robotics [352,353], robot-inspired path planning [354], and managing network traffic in computer networks [355]. Different artificial intelligence (AI) strategies have also been infused into manufacturing-based technologies, such as cybersecurity (Malware detection) [356,357] and augmented reality (AR) systems [358,359], to enable them to operate more autonomously and intelligently.

Another area where AI has played a useful role in the factory is in the prognostics and health management (PHM) of machinery [360]. The ability to detect deviations in the normal operating conditions of industrial components is helpful for the timely prediction, detection, and isolation of faults; and the prevention of costly and unplanned failures. Real-time visibility about the health status of individual machines and the entire production system provides immediate and long-term value to the production process. While the combination of data availability and the adoption of traditional ML techniques in recent years provided a lot of insights into component defects, root cause analysis, machine degradation, and remaining useful life (RUL), the complexity of the manufacturing systems has prompted a pivot to deep learning (DL) solutions, which are better able to handle the complexity of input data, providing hierarchical representations [361].

AI has also supported intelligent control of manufacturing systems. The growing complexity of controlled systems now means that no single control paradigm addresses the issues prompting the use of hybrid control systems that sometimes include both discrete event systems and continuous systems. To be effective, these hybrid control systems require intelligent control methodologies [362,363,364,365,366] to be embedded within them.

Optimal system configurations, performance evaluations, material flow modeling, throughput, etc., are all within the purview of AI. Most factories contain highly integrated systems comprising manufacturing cells, workstations, assembly lines, material handling systems, and a network of multiple machines, robot, and conveyors that support the flow of materials and their processing/refinement. These arrangements typically have finite buffer capacity. These finite buffers and periodic system failures of unreliable machines, and uncertainties and inter-dependencies that make factory operations nonlinear and stochastic [290,291], make the determination of optimal system configurations, performance evaluations, material flow modeling, etc., very challenging. A lot of work has been devoted to the analysis of the dynamics and performance of manufacturing systems [31,291,367,368,369]. Some of these, such as the queuing theory and Markov chain-based analytical modeling methods, have known limitations [367,369,370,371]. AI-based ML techniques appear to be compensating for these shortcomings [291,369,372,373,374,375].

AI solutions have also been used in job scheduling. To meet mass customization requirements, flexible manufacturing systems (FMS) are designed to easily adapt to changes in the type and quantity of products. A particular challenge that then arises during implementation is the job dispatching problem: several product orders could be awaiting processing within the same time window. Exact approaches (best for small-scale job dispatching problems) and heuristic-based methods (not very adaptive in highly dynamic environments) have been traditionally used to address these issues. As for their limitations, ML-based methods have been used to compensate for these deficiencies. Artificial intelligence applications will play important roles in the transition to *Factories of the Future*. More application examples are discussed below. These include a deep-learning-based quality control solution used for defect detection in the assembly line [376] and an AI-based (machine learning) and context-aware intrusion detection system [377]. Li, Bo-hu, et al. [26] performed an entire review on applications of artificial intelligence in intelligent manufacturing. Within industry, several manufacturers have also implemented industrial AI in their manufacturing ecosystems, moving from pilots to integration at scale. Referred to as the “lighthouse factories”, these organizations have realized significant financial and operational benefits in the process [378,379].

#### 8.2.4. Blockchain

This is a type of shareable ledger that runs atop a permissioned network [380]. It can also be described as a distributed network of nodes, typically running on multiple servers that feature a system of perpetually growing lists of trusted or verified asset transactions. Depending on the way the system is set up, trust is distributed across nodes within the network. These nodes are responsible for the verification, authentication, and integrity of block data before the ultimate inclusion of the new block into the growing chain. These continuously growing chain link of blocks is aptly referred to as a blockchain. Each block stores data associated with an asset (i.e., person, place, or thing). The relationship between the blocks is maintained through a mechanism that enables each block in the chain to inherit an immutable hash of the prior block that it is connected to. A sransaction within a blockchain is usually processed and stored without consultation with, approval of, or even the need for a central trust authority [381]. One of the main reasons blockchain is an important building block for a typical future factory is the fact that the data stored on it (e.g., transaction details) are immutable or remain unchanged, unaltered, and indelible, which means that the full history and data trails of all data, communications, and transactions are preserved, thereby establishing data provenance. A typical blockchain system includes a network of nodes that serves as a decentralized and trustless system which ensures data provenance and the efficient sharing of manufacturing (products and processes) information. It can be set up as a decentralized and connected network of manufacturing assets and computing nodes. This system will provide transparency, and audit the trail of assets and a third-party verification system of an organization’s manufacturing capacity, creating a mechanism for operationalizing “smart contracts” between different parties or entities. It will be essential for product customization and tracking of assets within a supply chain. Various factories and their value chains might need to be connected in the factory of the future, in what could be termed networked organizations and machines. In a typical network of the sort suggested, multiple third-party entities might be involved in various aspects of the production of a product, including receiving supplies (materials and spares), service provisioning, product design, fabrication, and production. Other aspects might include product testing (validation/verification), regulatory control, and shipment. Trustworthiness among the disparate parties (factories, distributors, suppliers, regulators, and other stakeholders) within such a network would be critical if all participants are to be at ease. Basic implementations of blockchain abound in the literature. Examples exist from healthcare [382], intellectual property protection of a 3D print supply chain [383], and machine-to-machine (M2M) interactions in the chemical industry where industrial plants trade electricity with each other over a blockchain. In a certain instance, a central server which typically manages information exchange and data authentication in an IoT system was replaced by blockchain (BC) technology, hence eliminating risks of device spoofing, false authentication, and several other security and privacy concerns [384]. Atin Angrish et al. [37] provided some great insight into blockchains with a focus on manufacturing.

#### 8.2.5. Mixed Reality

Mixed reality is a class of technologies that attempt to blend the physical and digital worlds. Mixed reality operates on a spectrum: virtual reality (VR) and augmented reality (AR) are the most prominent types. In this paper, we focus our attention on augmented reality (AR), mainly because it likely has the greatest potential to proliferate in the factory of the future. Augmented reality (AR) systems project context-sensitive [385] digital information in 2D or 3D forms (i.e., texts, stats, maps, videos, images, animations, characters, etc.) over real-life objects (the physical world) for the express purpose of providing additional information, context, instructions, or guidance about said objects, or processes that the objects are undergoing. They enable users to interact with real and synthetic elements of the real and virtual worlds simultaneously [386]. As a human–computer interaction tool, users (technicians, maintenance crew, etc.) can directly interact with the “extended” information to make informed decisions. Augmented reality (AR) systems complement (or augment) human abilities, providing users the guidance and support needed to complete tasks correctly in a consistent and efficient manner, leading to higher productivity, greater accuracy, and a marked reduction in expensive reworking by ensuring tasks are performed accurately, the first time. User-friendly and intuitive human interfaces alongside rich, appropriate, and context-aware content are factors critical for a great AR experience [387].

(a)**Types of Augmented Reality (AR) Systems:** There are several types of augmented reality (AR) systems, differentiated by application, functionality, or design. Of these, four main types stand out: (a) Marker-based AR: AR systems of this class display content (video, text, animation, 3D figures, etc.) on surfaces contingent with the detection of a predefined marker embedded on a static image (trigger photo) or a QR code, often using AR devices such as smart phones. (b) Markerless AR: Unlike Marker-based AR systems, they do not require physical markers for the overlay information to be triggered. They merely scan their environments to get their bearings and are generally guided by localization or positioning systems, such as GPS, accelerometers, and digital compasses. (c) Projection-based AR works just like typical projectors. They utilize image or video-based projection (with audio prompts, in some cases) to guide the pace, direction, and “every step” of a process. They help operators or factory workers through manual processes, enabling them to complete tasks quickly, efficiently, and consistently without recourse to hard-copy manuals and instructions. (d) Superimposition-based AR relies on the object recognition technique to first identify an object and then replace it or a portion thereof with an equivalent augmented image. An often-cited application of superimposition-based AR is in the medical field, where doctors sometimes superimpose live feeds (X-ray images) of a patient’s body part directly from an X-ray machine unto the patient’s actual body to better understand the internal condition of the body part.(i)*Hardware Devices:* Hardware devices are a necessary and integral part of augmented reality (AR) systems. There are several types of AR devices in common use. Some of these include handheld devices (HHD) [388,389,390] holographic displays, head-mounted displays (HMD), smart glasses/lenses and virtual retinal displays (VRDs), mobile phones (including smart phones) [390], wearable data-gloves [391], haptic devices [392], tablets, iPads, and computers.(ii)*Software Systems:* Software (or applications) also forms an important part of AR systems. Of particular interest are (a) tracking and registration algorithms and (b) development platforms (or content-creation applications). The primary function of the tracking and registration algorithms is the alignment of the two (real and virtual) environments or object categories. On the other hand, development platforms are the applications used for the creation of AR content. They include anything from low-level programming libraries to the more complex AR applications that integrate features for sensor data acquisition and integration, image, and audio rendering, and in some cases, even application engines.(b)**Industrial Applications of Augmented Reality:** Over the years, AR technology has continued to mature. Since it can simulate processes, augment tasks, provide remote assistance, enhance communication between teammates, and provide elaborate guidance to users, it has demonstrated relevance to manufacturing amidst the on-going re-imagination of the sector. There have been proposals [393], proof of concept studies, and actual applications [394] in a wide array of industries [48,52]. Its successful application at various manufacturing stages (planning, design [395,396], assembly [394,397], maintenance, etc.) is particularly notable. It has also found applications in different manufacturing processes and functions. A few of these are discussed below:(i)*Interactions with Process Information:* They have been used to digitally access and interact with procedural and process information, including IIOT related data [398] acquired in real-time, rather than relying on physical manuals and paper documents. Some have been used to display augmented 3D images, making it possible to view system components in multiple configurations, including exploded, cross-sectional, and internal views. For example, internal views come highly recommended for providing insights into the internal sections of opaque structures or systems where accessibility or worker safety is an issue [399].(ii)*Quality Control:* AR systems are already playing a huge role in automated real-time, in-production quality control. The mobile nature of most AR systems supports the relocation of the quality control functions away from static (fixed) input locations to mobile terminals, permitting intermediate inspections, and facilitating the flexible and cost-effective use of software license seats. The online, real-time, and decentralized characteristics of the AR systems provide the added advantage of instant access to and flexible flow of information to various manufacturing points where they are most needed. It also enables fast variance inspection, continuous, real-time error reporting, and documentation. Finally, the instant generation of enhanced quality assurance reports [400] immediately after the completion of each instance of an inspection routine [401] is not a possibility.(iii)*Process Support, Training, and Simulation:* AR technology has also been used to assist technicians and operators working on mechanical or technical tasks such as welding [402], machinery repair, and assembly operations, and even in controlling robots [403,404]. Some AR inspection systems incorporate features that provide graphic step-by-step instructions that can be used for process training. The step-wise design of these routines ensure that processes are performed in a consistent, accurate, and reliable manner. They can also be used as simulators for practice runs to help users develop and perfect their skills, ensuring that manufacturing tasks and processes are carried out right the first time. In the long-term, this helps with limiting errors and eliminating the need for rework. This level of expertise and dedication is useful for high-stress tasks where precision is critical.(iv)*Repair and Maintenance:* The repair and maintenance of complex machinery will be one of most consequential areas of AR application in manufacturing. Next generation AR-inspired maintenance systems are becoming important elements of the Factory of the Future [405]. They are now more often the products of the intersection of AI, IIoT, big data, and associated technologies and capabilities. Excellent condition monitoring, combined with dynamic predictive modeling, make for a successful predictive maintenance program. In the Factory of the Future, technicians going about in the normal course of their daily duties can be prompted by their wearable IAR devices (such as smart glasses or mobile devices) about “just-beginning” maintenance problems, way in advance of actual system or component failures. These AR systems not only detect and warn operators and technicians about these anomalies, but also offer on-the-spot visual analysis of the problem, display the service histories of the machinery, and deliver step-wise service instructions to aid in their resolution. For off-site maintenance, the AR systems can serve as remote collaboration tools where a technician can contact each other, collaborate with colleagues for resolving tough problems, or be remotely guided by a more experienced supervisor [406].(v)**Collaborative Product Design and Prototyping:** Almost all aspects of product design, for early to late stages, can now be collaboratively performed (end-to-end), streamlined, sped-up, and optimized using AR. These stages include ideation, conceptual design (encompassing generic functionality management [407]), preliminary design, the interactive generation of models or virtual product prototyping [397,406,408], design review, and evaluation [409]. Free-form surface generation features in some AR applications have been used to support easy creation of design alternatives and to enable parameter adjustments [407]. In automotive design, for example, AR-based design tools have been used to evaluate multiple interior design options by simply overlaying different photo-realistic 3D car interior mock-ups over real cars [408], eliminating the need for physical prototypes. AR-based design tools often generate sharable, high quality, 1:1 scale, photo-realistic 3D visualization of augmented design models that can be converted into AR compatible format and transmittable to stakeholder’s devices for easy and enhanced viewing. Availability and real-time remote access to these models make collaboration easy. They are enabling stakeholders (both designers and other collaborators, downstream in the product pipeline) to inspect and interact with the design models, and provide timely and objective feedback for design improvements, in advance of design approval and production [16]. The early detection of flaws facilitates design improvements and eliminates expensive post-production redesign costs. Several user-friendly, computer-aided AR design environments, such as ARCADE, are now available [410].(c)**The Challenge with current AR Systems:** Though there is growing interest in the use of augmented reality (AR) as a support tool across industry [411,412], one drawback of most AR-based maintenance systems is that most applications are currently passive and static in nature. They merely push information and provide no feedback mechanism capable of ingesting, analyzing, and looping back explicit and implicit user and environmental responses. A feedback system of this sort can enable the output of targeted information to users, continuous process refinement, and better tailored solutions. Examples of responses (data points) that can be routed back to through the feedback system include such data points as effectiveness of prior guidance, the experience of users, or even the user actions or inactions that could help preempt user intent. There is a need for more adaptive AR, with creative feedback loops that can actively engage users and help them to solve problems more creatively. Attempts have been made in the literature to spotlight this challenge and suggest creative ways of solving this problem [413].

There are several applications of mixed reality in manufacturing, as already alluded to. One of the earliest machinery maintenance and repair (laser-printer) mixed reality (MR) applications was developed by Steve Feiner’s team at Columbia University in 1993 [414]. Mixed reality (MR) has also been used to develop an application for improving performance in the execution of assembly tasks [415]. Applications of AR/MR for employee training and work on production lines [416], quality monitoring and inspection, maintenance, assembly, and safety have been reported [417,418]. Boeing once used mixed reality for cable harness design and the assembly of furniture, automobile door locks, and cockpit modules. This resulted in marked improvements in their assembly process [419,420,421]. This is in agreement with studies that show that mixed-reality-based techniques help reduce assembly process errors by as much as 80% [422].

#### 8.2.6. 3D Printing (Additive Manufacturing:)

3D printing (also referred to as additive manufacturing (AM)) has been referred to as one of the major technologies of the 4th Industrial revolution [423] because of its potential for massive disruption of the status quo across many industrial sectors [424]. Alongside other disruptive technologies, such as IoT, cloud, big data/analytics, and AI, it is expected that AM will create the necessary conditions for expedited processing, rapid prototyping, customized production, and agile manufacturing. Additive manufacturing has various definitions, but one of most descriptive is the one captured in the ISO/ASTM 52900 standard [425], which defines it as the “process of joining materials to make parts from 3D model data, usually layer by layer, as opposed to subtractive manufacturing and formative manufacturing methodologies”. Irrespective of what definition is accepted, the general principle upon which this technology rests is the creation of 3D geometries through the precise addition of basic building blocks, such as grains of powder or polymer filaments, laid out as a series of cross sections, often layer by layer. This occurs with minimal material waste and a nominal or limited need for post-processing [426]. The creation of these 3D geometries is driven by digital instructions (geometric information) typically sent from a computer (CAD model) to a printing head, nozzle, or related printing technology. The instructions which are processed as points, lines, or areas help guide the direction and rate of material deposition [427]. Over time, the technology has continuously matured, getting to a point where it is now feasible to work with all sorts of materials, including ABS plastic, photo-polymers, stereo-lithography materials (epoxy resins), metals (e.g., steel, titanium), wax, and even biological materials. There are seven categories of additive manufacturing (AM) [428]. Each category includes several techniques, some of which are well known within the AM community. The processes involved in the techniques differ depending on the materials used and the mechanisms employed. They include (a) VAT photo-polymerization, (b) material jetting (e.g., continuous on demand (CoD) and drop on demand (DoD)), (c) binder Jetting, (d) material extrusion (fuse deposition modeling (FDM)), (e) powder bed fusion (e.g., direct metal laser sintering (DMLS), electron beam melting (EBM), selective heat sintering (SHS), selective laser melting (SLM), and selective laser sintering (SLS)), (f) sheet lamination (ultrasonic additive manufacturing (UAM) and laminated object manufacturing (LOM)), and (g) directed energy deposition (e.g., laser engineered net shaping, directed light fabrication, direct metal deposition, 3D laser cladding).

Traditional manufacturing is generally expensive due to the high labor cost and complicated set-up (machinery and process) requirements. Re-purposing and switching product lines can take weeks, or even months. This contrasts with additive manufacturing, which lends itself to quicker process adjustments and easier adaptation within a larger production line. Furthermore, changing production speed or switching between products can be easy and quick, and relatively fewer operational staff are often required to achieve equivalent work outputs. Additive manufacturing (AM) has the potential to democratize manufacturing on a global scale. It is cheaper and best suited for high-value, low-volume, small, and short-run parts production. AM is well positioned to benefit the growing demand for product personalization and mass customization. They are also very useful for creation of parts and structures with complex geometries and require flexible designs. Examples can be seen in machine parts, dental work (precise crowns and dentures), artwork, customizable gifts, etc. In the live sciences, layers of living cells have been printed over one another to create human skin. Some of these can potentially be surgically implanted into other living materials to fix complications on the bodies of burn victims. They are also being evaluated for skin product testing, to help reduce the controversial use of live animals for biological tests. There is also an emerging consensus amongst researchers that at some future time in the future, it will be possible to print human organs and increase options for people on organ waiting lists.

In manufacturing, the fabrication of tools necessary for producing parts or components is important. Agile tooling, a specific tooling approach which involves the efficient and cost-effective design and fabrication of tools such as molds, patterns, dies, jigs, and fixtures, is an important aspect of manufacturing-related tooling. The most popular agile tooling techniques include die-casting, die-stamping, hydroforming, and thermoforming. Some of these tooling has been produced at great turn-around times and at a fraction of their regular costs, thanks to additive manufacturing (3D printing) processes. For example, tools, typically created with the vacuum forming process, can now be produced faster, more efficiently, and at lower costs using fused filament fabrication (FFM), an additive manufacturing process. The ability to produce these tools faster would mean quicker prototyping and shorter time to market. With the adoption of additive manufacturing (AM), a Future Factory (central hub) can potentially fill a demand for a replacement part initiated by a customer, from a thousand miles away, in record time, without any shipping requirements. An example of how this can play out is that a virtual 3D model of the requested part, released by the Future Factory to a cloud location, becomes immediately accessible to an authorized, out-sourced third-party 3D print location, closer to the customer. Following a few clicks, the 3D model is printed and becomes available for immediate customer pick-up. This arrangement will reduce wait times, accelerate production rates, and eliminate shipping costs. It will also reduce time-to-market, making the creation of products cheaper and more accessible.

Opportunities to combine evolutionary or genetic algorithms with 3D printing to speed up design and determine parts with the best configurations for specific industrial service are very numerous. Researchers from NYU [429] used genetic algorithms alongside 3D printing to determine the ideal wing shape for a fast-flapping flight. Mitra Asadi-Eydivand et al. [430] also used 3D and evolutionary algorithms to determine the optimal design of a scaffold. One of the biggest challenges that pervasive 3D printing will face within the industrial space are the issues of the ownership and control of intellectual property. For example, how will the owner of a design or electronic product specifications be compensated once the files end up in the hands of a client? How can the distribution of those files, once out of the owner’s control, be monitored? Some proponents have suggested online marketplaces, brokers, or clearing houses that would serve the dual purposes of regulating access to electronic specifications and managing compensations and payments to intellectual property owners.

Applications of additive manufacturing (3D printing) are beginning to have real-life impacts in a number of industries, including aerospace and defense (A&D), medicine, the automotive industry, bio-medicine, etc. It has been used for manufacturing tooling (e.g., hand-tools for testing and assembly), light weight or highly complex parts/components, spare parts, and functional prototypes. When used for complex parts, it is often the case that only low volumes of the parts are required, and it is impractical or uneconomic to use traditional manufacturing methods. Due to its advantages, such as the capacity for high precision fabrication of complex geometry [431], the potential for significant reductions in material waste (material efficiency), and flexibility in design, it has been an ideal solution for fabricating light-weight parts in the aerospace industry. It is also used for ensuring availability of spare parts [432] and the maintenance and repair of aerospace parts such as wings, turbine blades, rocket parts, and other sophisticated components. It was used to fabricate a complex injector head for the Ariane 6 launcher using selective laser melting (SLM) technology and a nickel-based alloy. The Ariane 6 launcher was developed Ariane Group, a joint venture between Safran and Airbus Group [433]. A number of manufacturers (e.g., Ford) use 3D printing for fabricating tooling. AUDI (in conjunction with SLM Solution Group AG) fabricated prototypes and spare parts using 3D printing [434]. Many other examples of the applications of 3D printing to save costs, speed up production, or execute difficult tasks abound in the open literature and industry

#### 8.2.7. Autonomous Robotics

The factory of the future will have two employees: humans and robots. Of the two, robots are expected to play a prominent role because of the high likelihood of a disproportionate reliance on machines for industrial tasks as compared to humans. The word “robot” is a Czech word that is literally interpreted “forced labor”. They are very useful for tasks that require high levels of accuracy or are dangerous or repetitive. They bring to tasks such benefits as improved effectiveness, higher efficiency, and reliability. They can perform tasks that humans cannot or should not (e.g., demeaning or dangerous tasks). Aside from augmenting human efforts, they also create the flexibility for under-utilized labor to be replaced or re-assigned. Long term, they are relatively more cost effective and facilitate higher productivity. There are different classes of robots, including commercial robots, industrial robots, and service robots. The focus of this section is industrial robots. *Industrial Robots:* ISO 8373:2012 defines an industrial robot as “an automatically controlled, reprogrammable, multipurpose manipulator, programmable in three or more axes, which can be either fixed or mobile for use in industrial automation application”. The ability of Industrial robots to perform high-precision work accurately, repeatably, and quickly is helping factories deliver high quality products and driving plant efficiency and profitability. Industrial robots will become ubiquitous across most manufacturing environments. As these already highly productive robots are becoming AI enabled (AI robots) and fully integrated into the data-rich manufacturing ecosystems, both sharing and receiving data/information with other subsystems, it is easy to see why they will eventually become the workhorses of the factory of the future. Over time, it is expected that AI will morph into more intelligent systems with strong cognitive abilities. Robots are used for a variety of tasks within the manufacturing space. Some of these include mechanical cutting, grinding, deburring, polishing, welding (i.e., arc welding, spot welding, etc.), and painting. Other common robot tasks include picking, packing and palletizing, material handling, assembly, firefighting, and patrolling of warehouses and storage areas.

Based on one classification, three main types of robots operate within industrial environments. These include: (a) traditional robots, (b) collaborative robots (cobots) and (c) mobile robots.

(a)*Traditional Robots:* The technologies underpinning traditional robots are generally more mature. They generally have high payloads, have longer reaches, and are able to achieve very high efficiency levels, even at high production speeds.(b)*Collaborative Robots (Cobots):* ISO 10218-2 defined cobots as robots designed for direct interaction with a human within a defined collaborative workspace. Workspace refers to the safeguarded space where the robot and a human can perform tasks simultaneously during production operation. Generally, they are relatively easier to program, enable more efficient production adjustments, and can more flexibly adapt to new requirements than traditional robots. For implementation, they require minimal changes to existing production layout and can be easily redeployed for different tasks, as necessary. A defining characteristic of these robots is that they work collaboratively with human workers, without concerns for worker safety. They possess several integrated safety features, including collision detection technologies, minimized pinch points, safety-rated monitored stops, and well controlled force and speed. Human workers can focus on tasks that require strong cognitive abilities, whereas the robots can be assigned repetitive tasks and other activities that require precision or heavy lifting. Robots that work alongside humans are referred to as cobots.(c)*Mobile Robots:* Mobile robots have a general awareness of their environment and the ability to effectively navigate through it in the process of accomplishing assigned tasks. While traditional robots are usually stationed at fixed locations and are mostly assigned tasks that do not require a lot of flexibility. Mobile robots, on the other hand, are usually ambulatory and they are best suited for constantly changing factory environments. Using their navigation systems, they transverse entire factory floors autonomously, seamlessly integrating themselves into the manufacturing ecosystem. They can stop, move, slow down, or navigate away from obstacles using sensory information obtained from a wide array of localization and navigation sensors embedded within their bodies or attached to their surfaces. Two main classes of robot sensors exist, i.e., exteroceptive and proprioceptive sensors. Exteroceptive sensors help the robots discern and understand their environments. Examples of exteroceptive sensors include stereo cameras, pan/tilt/zoom cameras, lasers and 3D lidar systems, projection-based systems, audio/video feedback systems, touch sensors (whiskers or bump sensors), and GPS, proximity, and certified safety sensors. Then there are proprioceptive sensors, which are sensors that gather information about the robot itself. Examples of proprioceptive sensors include accelerometers, gyroscopes, magnetometers, compasses, wheel encoders, and temperature sensors). These sensors, alongside accompanying algorithms, enable the mobile robots to both understand and safely navigate their environments. For this reason, they are very safe to deploy alongside human workers, with whom they sometimes work collaboratively, transforming them from mere machines to fellow workers. The basic idea of the mobile robot is essentially moving the robot to the work instead of moving the work to the robot. Mobile robots would best benefit such tasks as automated assembly, inspection, painting, or welding of huge industrial components, such as airplane frames, large engines, and giant offshore or space structures. Due to their large sizes, working on such components with two or three stationary robots can be inadequate because of the limitations on the reaches of such robots. Alternative courses of action could be to either add more robots (a costly option) or employ mobile robots which are not limited by reach due to their ability to move around the entire structure. Compared to traditional robots, mobile robots are more flexible and adaptable. Their ability to maneuver through space and structures helps shorten throughput times, improving efficiency and cutting down on production time. Mobile robots have a variety of locomotion mechanisms [435,436], e.g., flying (drones) [437], rolling, walking (legged), swimming or water-based movement (underwater vehicle manipulator system) [438], crawling, moving on tracks, and using propellers. Automated guided vehicles or automatic guided vehicles (AGVs) are amongst the most common mobile robots within the manufacturing industry today. Additionally, they are poised to become even more ubiquitous as adoption continues to grow. They are currently used for moving materials, supplies, and products around manufacturing facilities. Unmanned aerial vehicles (such as the drone) are the next set of robots that will grow in relevance within manufacturing. They would be especially useful for picking up and dropping items, and product and quality inspections, especially the inspection of equipment or machinery at hard-to-reach locations (e.g., high elevation or dangerous locations) using thermal and video cameras.

## 9. Discussion: Recommendations and Future Research Directions

Following our review of the literature, the dominant theme is the general idea that the *Factory of the Future* is about data and the value that can be extracted from it to improve operations and optimally create value. It is about networked or connected assets (machines, people, devices, networks, etc.), and the different communication technologies and protocols that enable them talk to each other (communicate seamlessly). As a large information network, the *Factory of the Future*, is about capturing raw data from said network of connected assets (plus related manufacturing processes) and the transformation of the acquired data into actionable intelligence [439,440], with a view of facilitating decision making, optimizing manufacturing processes, sustainably producing goods and services, and improving overall performance. Amongst many important qualities, intelligence and cognition are machine characteristics that make a factory smart. They empower the factory with the ability to know, understand, interpret (correctly), and respond (effectively) to events happening within and around it. Both characteristics can be developed if several challenges “slow-walking” progress towards the full realization of the *Factory of the Future* are addressed. Given that the *Factory of the Future* is a huge information network, most of these problems are somehow related to processes (such as the collection, storage, analysis, security, or analysis of data) or the protocols, frameworks, or technologies used to manage the said processes. Some of these will be discussed below.

### 9.1. Communication Protocols and Technologies

The *Factory of the Future* contains a huge collection of networked assets. One of the big challenges facing the discipline is how to make each of them “speak the same language” and seamlessly communicate (back-and-forth) with one other. The emergence of the Industrial Internet of Things (IIoT) [441] and open communication standards/platforms and advanced advanced M2M communication protocols, such as the Open Platform Communication Unified Architecture (OPC UA) [219,442,443] and MQ Telemetry Transport (MQTT), has advanced data collection and synchronization efforts. A lot of research still needs to go into developing new and improved communication standards and framework. There is also room for developing protocols that are nimbler and easier to configure.

### 9.2. Digital Infrastructure

To become a completely connected system, the *Factory of the Future* has to be fully integrated into extensively networked manufacturing supply chains. This would require data-driven architectures capable of linking, at high fidelity and accuracy, all data generated at every stage of the manufacturing process and product life cycle. Currently, there are gaps in the virtual infrastructure. A lot of research is required for building out systems, architectures, and technologies that help interconnect different entities within the manufacturing network, e.g., Factory-to-Factory communication.

### 9.3. Data Collection and Characterization

Data collection methods that are better linked or aligned with domain knowledge need to be developed. It is also important to realize that for data to be valuable, its context must be understood. Further manufacturing domain-based research in data contextualization and linking methods, such as graph theory, category theory, and linked data, are critical. The eventual democratization of these methods will also serve the industry well. Determining relationships between terms and concepts and extracting value from these relationships should be further explored by domesticating research into semantic indexing techniques. Frameworks that better connect concepts are required for more productive analytics of multivariate data, especially at the network or system level. Overall, multi-disciplinary research in natural language processing (NLP) will greatly benefit the manufacturing domain as we try to make sense of all the data we collect.

### 9.4. Virtualization

A major priority of the *Factory of the Future* is the seamless exchange of data between all nodes within the manufacturing network in real time or near real-time. This will require a level of decentralization that would make the hierarchical automation pyramid obsolete. Virtualization of assets and the production process using technologies such as digital twins, the asset administration shell (AAS) and data distribution services (DDS) will enable bidirectional relations between physical assets and their cyber twins. The communication pipeline between these twins will enable optimization, real-time remote monitoring, easy and early identification of failures and inefficiencies, optimization of production processes, continuous improvement through simulations, etc. Notwithstanding the progress that has already been made with the virtualization of assets, there are still a lot of challenges yet to be addressed that will require some research work (basic and applied). One such challenge is that of insufficient synchronization capabilities. In many cases, simulation models are not (fully) accurate due to synchronization issues [444]. Simulation models should not be distinguishable from their physical counterparts. There are also issues around the prediction of the state of complex systems and the complexities around big data collection, storage, and processing [16,445]. These challenges will require further research work. The development of dedicated conceptual frameworks and reference models for digital twins would also be helpful [444].

### 9.5. Interoperability

Interoperability and standards are a twin challenge affecting the realization of the *Factory of the Future*. The proliferation of technologies, protocols, standards, and frameworks has increased the need for interoperability at different levels and among different aspects of the manufacturing ecosystem. Due to the lack of standardization early in the emergence of Industry 4.0, many manufacturers, trade associations, and even nation states developed proprietary solutions in the hope of taking advantage of the prime mover advantage and possibly monopolizing the market. No players have been able to dominate any major aspects of the industry. Contrary to this, a measure of controlled chaos has ensured, which is currently exacting cost and speed penalties on the industry, making it difficult for the digitization process to proceed at scale. Several standards organizations, trade grounds, and industry/government-led alliance initiatives have emerged to address some of the challenges. The various types of interoperability are discussed in the literature [160]. There is a need for interoperability on various fronts, including physical assets, in the transfer and exchange of data, in the physical and functional architecture, etc. A lot of effort is being put into developing standards and promoting interoperability. It would take some time for some of the handwork to really pay off. Some of the most common smart manufacturing architectures, reference models, and standards, along with the on-going efforts, are widely available in the literature [177,446]. Currently, there appears to be a haphazard approach to the development of solutions, and since most of the architectures, frameworks, and reference models are still in the conceptual stages, their real-world implementations (specific case studies) would be a significant way to contribute to the literature and the discipline.

### 9.6. IT/OT Security

Due to the IT–OT (information technology to operations technology) integration and the proliferation of connected assets characteristic of the *Factory of the Future*, there are major concerns about cybersecurity risks in smart manufacturing. The risks will only grow. Each additional device introduces some extra level of marginal vulnerability, a situation that collectively increases the risk of cyber-related compromise due to the growing number of vulnerable endpoints. Known attacks have not only been directed at passive devices—sensors, actuators etc.—but have also hit edge devices, PLCs, HMIs, and control systems used to manage industrial operations. Beyond the known static attack vectors, the number of stakeholders (employees, contractors, vendors, suppliers, clients, etc.) that have access to the connected OT systems has also multiplied the potential vulnerabilities of the factory. Many cyber-attacks have been reported within industrial and manufacturing facilities in the recent past. In one case, hackers were able to gain control of the main network of a German steel factory [447], through their corporate endpoints. In a different instance, using a watering hole attack, the Remote Access Trojan (RAT) was used to compromise industrial control systems with the energy sector [448]. There have also been cases of the modification of the operation of physical assets. An example is the Stuxnet [449] attack that targeted physical assets at an Iranian nuclear facility. This computer worm specifically targeted programmable logic controllers (PLCs), used to automate electromechanical processes. It was reported to have even modified the operation of the connected motors by altering their rotational speed. There was also the case of the suspected ‘logic bomb’ that exploded the Trans-Siberian Pipeline [450]. The reports are almost endless. There have been several research endeavors directed at addressing vulnerabilities in supervisory control and data acquisition (SCADA) networks [451,452,453,454]. Several research works have also been directed to addressing intrusion detection issues in programmable logic controllers (PLCs) [455,456,457,458,459]. While there is a patch work of activities meant to deter these attacks, there are several research opportunities in the cybersecurity of the *Factory of the Future* that need to be explored to address the challenges holistically. For one, there is a need to develop secured standards of communication that leverage the Internet as a gateway to bridge the IT–OT divide. There is also potential for research work that promotes default cybersecurity protections (i.e., secure by default and cybersecurity by design) in industrial architecture, frameworks, and software/hardware solutions. As traditional threat detection techniques and modalities are inadequate for addressing the threats within the smart manufacturing domain, there is also a need to develop more advanced AI-enabled threat detection and security analysis systems that can help detect vulnerabilities and respond autonomously. The need for personnel with the right mix of skills has not been greater. As part of measured directed at addressing the skills gap, curricula directed at addressing the security of cyber-physical systems have been implemented [460,461,462]. More work clearly needs to be done in this area. Experts with requisite skills in the fields of information security and industrial control systems/production processes would be central to the effort to mitigating the threats.

The *Factory of the Future* is a data-rich environment that harnesses intelligence from multiple information streams, i.e., assets (including people), processes, and subsystems, to help create value, new forms of production efficiency and flexibility. Being able to realize the goals of next-generation manufacturing will depend on solving a lot of the challenges with the infrastructure (physical and digital) and processes required to protect the integrity of data and optimally extract value from it. While the traditional factory relied on automation, the main goals in the development of next-generation factories are to anticipate uncertainty and to ensure that the factory and its many sub-systems will be able to correctly interpret and disruptions and respond appropriately with little or no human intervention. The growth of artificial intelligence, and particularly the progress in the field of natural language processing (NLP), have made it possible to analyze data, extract meaning from them, and determine appropriate actionable insights, all with no or limited human supervision. Though the current advancements in manufacturing technology require massive investments, the long term value (cost reduction, speed to market, mass customization, product personalization, etc.) will most certainly be worth the time and investment.

## 10. Implementations of the Factories of the Future

While there are still many challenges on the path of the full realization of the *Factory of the Future* and the long-anticipated Fourth Industrial Revolution, many organizations (academia, industry alliances, and companies) are already making massive strides in that direction. Academia is very engaged with training and research in smart manufacturing solutions. Event test beds for solution development and analysis are in use. Similar efforts are underway in various research centers all over the world. Industry alliances are also helping crystallize and achieve the smart manufacturing ideals and aspirations of their corporate members. Some discussion on these fronts is given below.

### 10.1. Industry and Implementation of Smart Manufacturing Solutions

Within industry, many companies have been making transformative strides through the digitization and virtualization of different aspects of their manufacturing processes. Factories that have morphed into smart manufacturing systems with a full coupling of all their subsystems are rare but remain the end goal. The most visible discussion about industrial applications of smart manufacturing that has resulted in the realization of real value centers was around the World Economic Forum’s Global Lighthouse Network, which is a a platform to used to promote the development, replication, and scaling of smart manufacturing solutions. It is comprised of companies (world-over) that have led the pack in demonstrating the value (growth, productivity, resilience, and environmental sustainability) in transitioning to smart manufacturing. As at 2022, 103 manufacturing Lighthouses have been identified, across the globe, from different industries. Examples of some of the companies that have made the list include: Baoshan Iron and Steel (Shanghai, China), AGCO (Marktoberdorf, Germany), Hitachi (Hitachi, Japan), Johnson and Johnson DePuy Synthes (Suzhou, China), Micron (Singapore), Weichai (Weifang, China), Henkel (Düsseldorf, Germany), Petkim (Izmir, Turkey), Johnson and Johnson Vision Care (Jacksonville, USA), and Groupe Renault (Curitiba, Brazil).

### 10.2. Industry Alliances and Research Institutes

National institutes/laboratories (such as the National Institute of Standards and Technology (NIST), Platform Industrie, CESMII, the United States’ National Institute on Smart Manufacturing) and Industry alliances (such as the Industry IoT Consortium and Taiwan Association of Machinery Industry (TAM)) are all examples of institutions working to advance smart manufacturing goals. Some of these have test-beds that have contributed significantly to doing the required pre-work needed, in a solution-agnostic manner, before industry can scale the solutions. There are several smart manufacturing test-bed implementations reported in the literature for different aspects of the manufacturing process or operation [463,464,465,466,467,468,469,470]. The Industry IoT Consortium typically has several test-beds [471] at any one time working on next-generation smart manufacturing solutions. Some specific examples of their test beds include: the Smart Factory Machine Learning for Predictive Maintenance Test-bed, the Smart Manufacturing Connectivity for Brown-Field Sensors, the Smart Printing Factory Testbed, the Time-Sensitive Networks Test-bed, the Track and Trace Test-bed, etc.

### 10.3. Smart Manufacturing in Academia

There is a lot of research (smart manufacturing-based) currently going on within academia globally. The future work force is also being trained and equipped (sometimes with real-life learning factories): for pilots, many test beds have been deployed for the evaluation real-life solutions. Test-beds create platforms for researchers and practitioners to develop innovative smart manufacturing solutions in a flexible and effective way. The System-level Manufacturing and Automation Research Testbed (SMART) at the University of Michigan [472] and the University of Sheffield AMRC, Factory 2050 (UK) are but two examples. The “neXt Future Factory” laboratory at the University of South Carolina is part of the global value chain dedicated to advancing smart manufacturing. As a research and development concern, the *neXt Future Factory Lab* located within the McNair Aerospace Center is the center of gravity for advanced manufacturing at the University of South Carolina (UofSC). The laboratory has a two-pronged approach that focuses first on research and development (R&D) and secondly on industry engagement. On the R&D level, the laboratory is focused on determining the most optimal approaches for connecting different manufacturing modules and enabling them to exchange information efficiently, reliably, and quickly (Interoperability). Another issue of interest is to figure out efficient ways of connecting these modules securely and reliably to the Internet (connectivity and cybersecurity). A third issue is how to effectively collect, manage, and analyze disparate manufacturing datasets so that business intelligence can be gleaned from them in real-time. This is important because we believe that data hold the necessary intelligence to influence technology innovation, competitiveness, and productivity growth in the manufacturing business. On the industry engagement vertical, our laboratory provides industry a secure and advanced platform with which to investigate solutions, and build proof of concept solutions and MVPs alongside our researchers and students, thereby de-risking their digital transformation projects and improving product and process efficiency.

#### 10.3.1. The “neXt Future Factory” Test Bed

Figure 22 shows the *Future Factory* test bed at University of South Carolina. This manufacturing test-bed (or functional mini-factory) is designed to support the study, exploration, maturation, and exploitation of manufacturing control elements, communication protocols, and a variety of emerging technologies with a view to advancing smart manufacturing (and related) objectives. The test bed is technology agnostic and relies on open standards for all its communication and data modeling requirements. It also has a flexible implementation architecture that makes it possible for new technology to be easily integrated and tested. This is to support the testing and maturation of new technologies and data strategies through industry use cases. It also supports the rapid creation of different iterations of product layouts to cater to the needs of diverse industrial products. The test-bed consists of several Yaskawa robotic arms, conveyor belt systems, and an array of multi-vendor and multi-platform equipment and devices, including programmable logic controllers (PLCs), edge devices, human–machine interfaces (HMIs), cameras (FLIR, infrared, thermal), sensors (wired and wireless), variable frequency drives (VFDs), actuators, etc. The various modules of the test bed are connected to the Internet using the 4LTE and 5G networks. Data transfer between automation devices within the test bed, and between the test bed and some sister sites as part of our Factory-to-Factory project, is made possible through the IEC 62541 standard OPC UA. The platform also has access to a multi-vendor array of software applications that enable the real-time collection, ingestion, and visualization of data, the implementation of advanced process simulations solutions, path planning, offline robot arm programming, the development of visual inspection, AR/VR, digital twin solutions, etc. In the future, we hope to be able to use the test-bed as a DevOps platform for streamlining all development and operational processes, including coding, analytics, modeling, deployment, updates, etc.

#### 10.3.2. Collaborations

The laboratory has multiple academic and governmental agency partners that work together to advance various educational and policy goals. Besides providing our researchers a forum to investigate different IT/OT phenomena and affording our students practical training opportunities, the facility also showcases the opportunities inherent in digitizing manufacturing to help drive productivity, reduce defects, cut cost, and accelerate time-to-market. We also work closely with some of the most innovative companies and consultancies in the country, each bringing in their best minds and an interesting amalgam of advanced technologies offerings (hardware and applications) to help address some of the most challenging problems in the advanced manufacturing space. The test-bed provides these industrial partners the platform to create different technology configurations, deploy them in real-time, and explore their efficiencies without the possibility of disrupting production, which helps them essentially de-risk the development of novel digital solutions. Besides being a showcase for the technologies of partner companies, the laboratory also performs applied research and development for industry. All of these activities have created interest within industry and driven different forms of business value, including enabling strong customer engagement, in addition to interest in our scholarship and our students (for employment), who are active participants in most of the R&D work.

## 11. Summary and Conclusions

This paper reviewed the meaning, essence, characteristics, and applications of the *Factory of the Future*, alongside its key supporting technologies. As an Industry-4.0-enabled manufacturing system, the *Factory of the Future* is flexible, adaptive, and transparent and is expected to replace the current (traditional) manufacturing system. The wide adoption of the *Factory of the Future* and its ideals is expected to transform the traditional factory from merely housing mechanical operations into an ultra-high-tech system that uses innovative smart technologies to improve production processes, reduce cost, encourage mass customization, and enable the prediction of potential problems even before they occur using predictive maintenance/analytics solutions. It will also support the efficient tracking of assets at each stage of the supply chain to improve asset visibility, control, and insight, enabling advances in inventory management and supporting improvements in logistics. The paradigm shift is clearly about making the manufacturing infrastructure and its supporting ecosystems smarter, quicker, and nimbler, while using data as a critical asset for self-optimization, self-adaptation, and competitive intelligence. The ability to ensure that assets are connected and that they “talk to and learn from each other” seamlessly is a defining feature of the Factory of the Future and is made possible through the deployment of machinery, components, and technologies in a manner that ensures their interoperability and connectivity. Even though significant advances have already occurred in recent years, the goal of a fully digitized and highly networked manufacturing sector remains. Many solutions are still in their early stages of development (aspirational, conceptual, pilot, testbed, etc.). It is, however, noteworthy that a lot of progress is being made at the policy, research and development (R&D), and implementation levels. As new technologies and systems become available, it is critical for researchers and practitioners to continue to push the technological envelopes to ensure the full transformation of significant parts of the manufacturing ecosystem. There is also the urgent need for all stakeholders, including the governments and the organized private sector, to help with scaling some of the proven solutions and ensure rapid industrial diffusion beyond the walls of research facilities, test-beds, and well-funded factories.

## Figures and Tables

**Figure 1 sensors-22-05834-f001:**
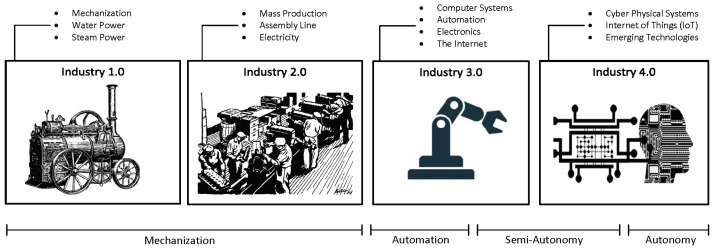
Diagram depicting the four industrial revolutions and degrees of system sovereignty.

**Figure 2 sensors-22-05834-f002:**
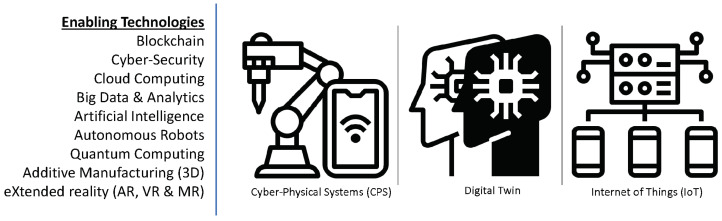
Emerging technologies enabling the development of the Future Factory.

**Figure 3 sensors-22-05834-f003:**
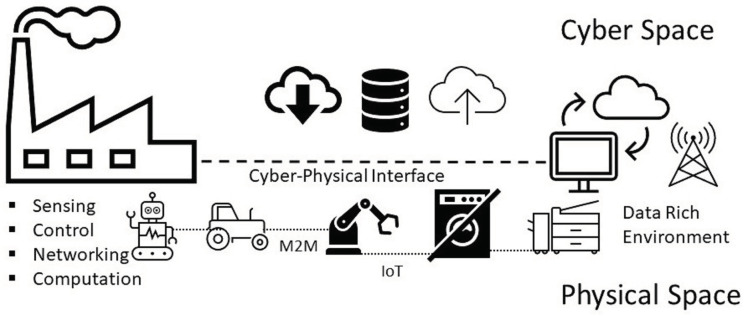
Cyber Manufacturing Systems (CMS).

**Figure 4 sensors-22-05834-f004:**
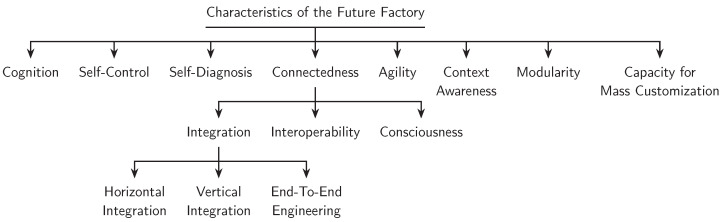
Characteristics of the Future Factory.

**Figure 5 sensors-22-05834-f005:**
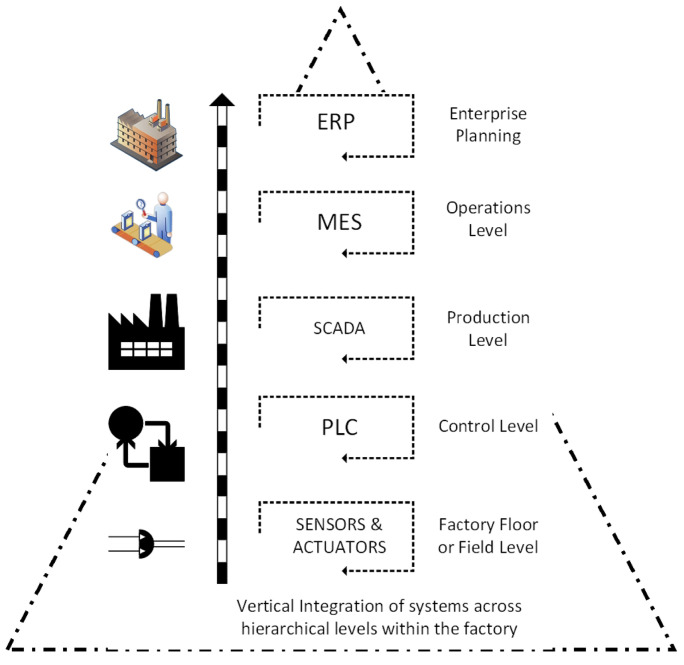
Automation pyramid.

**Figure 6 sensors-22-05834-f006:**
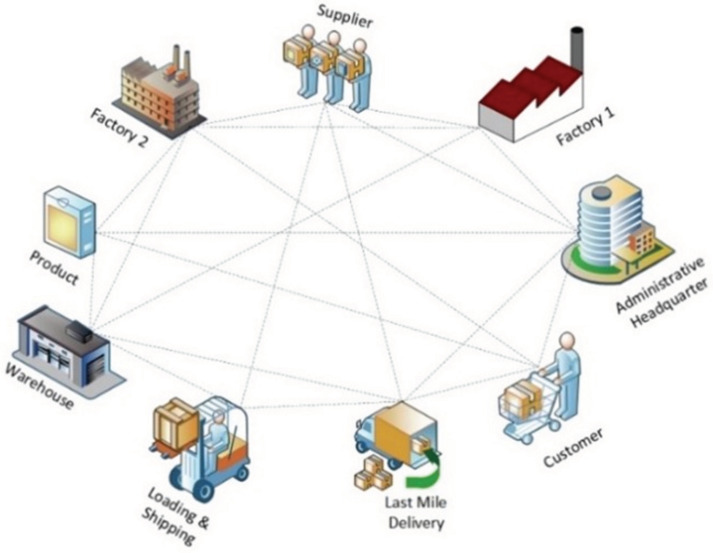
Horizontal Integration.

**Figure 7 sensors-22-05834-f007:**
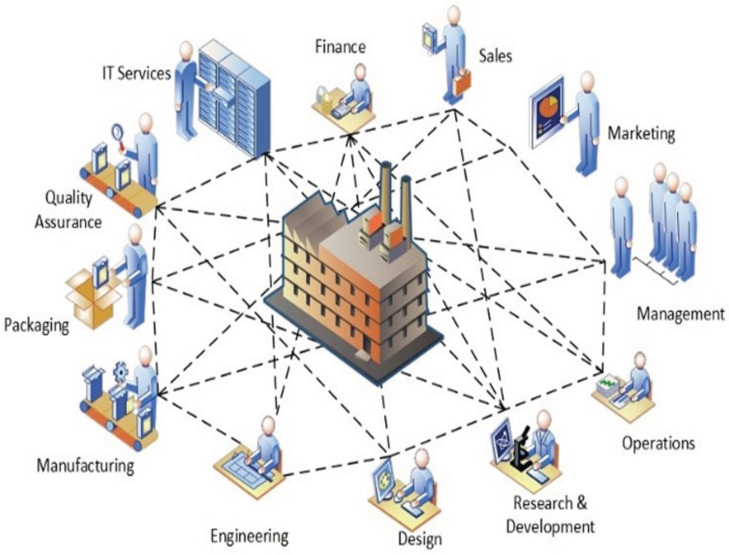
End-to-end integration.

**Figure 8 sensors-22-05834-f008:**
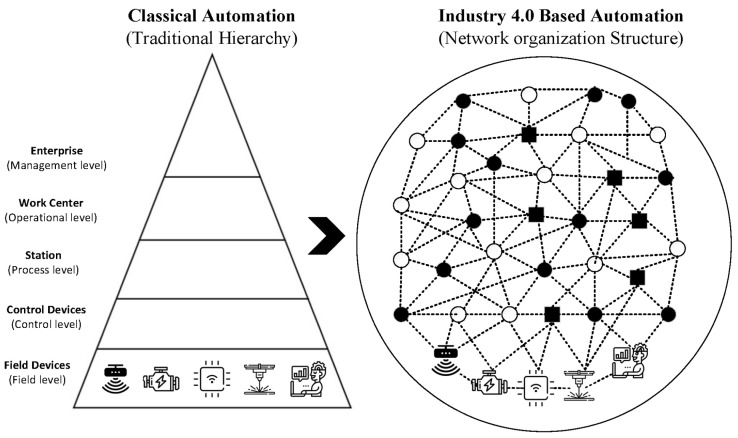
Classical automation pyramid laid side by side with a CPS-based automation model.

**Figure 9 sensors-22-05834-f009:**
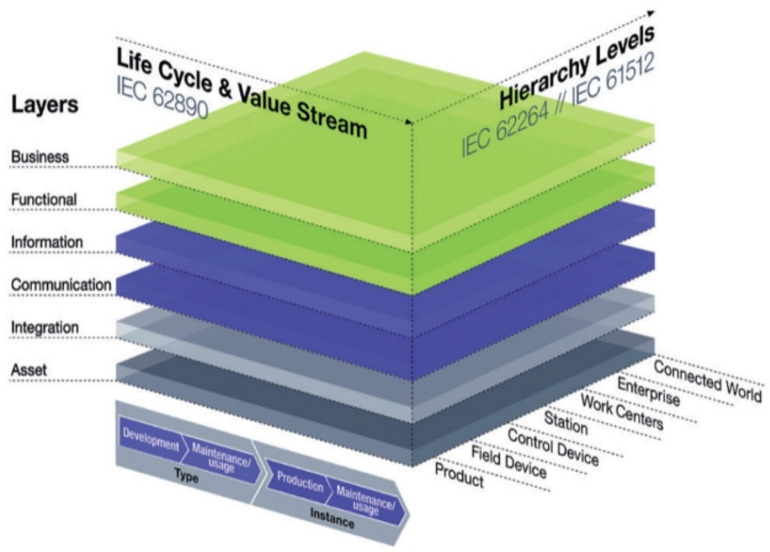
Reference Architectural Model Industrie 4.0 (RAMI 4.0). *Source:* Plattform Industrie 4.0.

**Figure 10 sensors-22-05834-f010:**
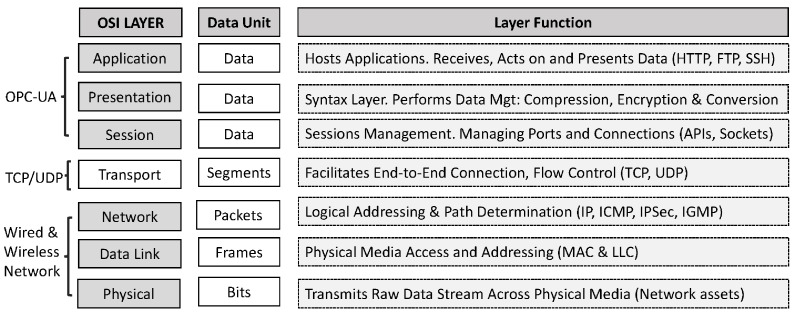
The seven OSI layers.

**Figure 11 sensors-22-05834-f011:**
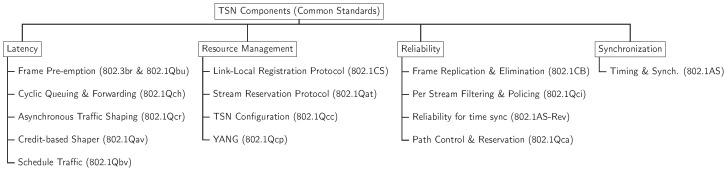
TSN components.

**Figure 12 sensors-22-05834-f012:**
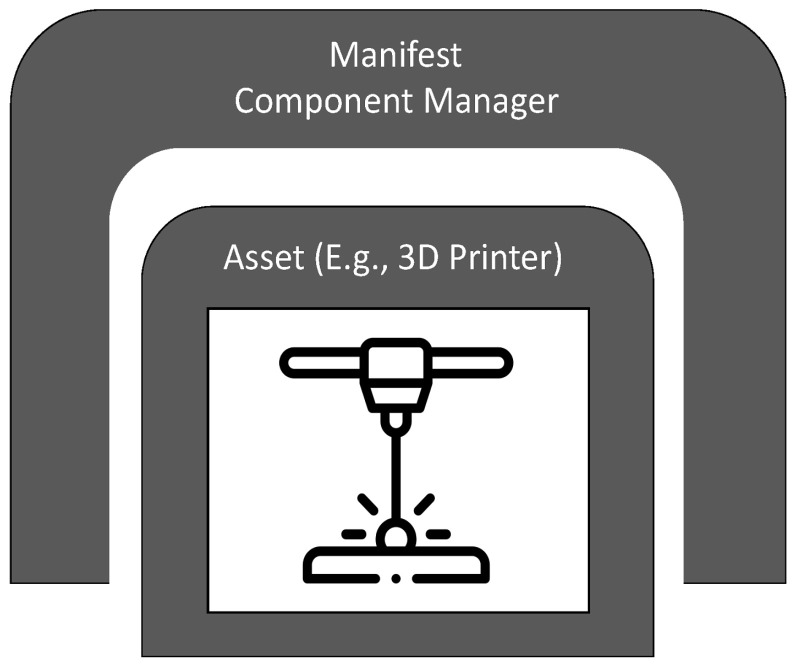
AAS showing an asset (3D Printer).

**Figure 13 sensors-22-05834-f013:**
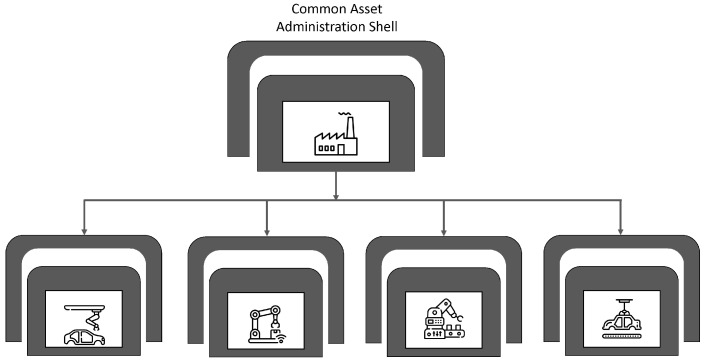
Common intra-factory asset administration shell.

**Figure 14 sensors-22-05834-f014:**
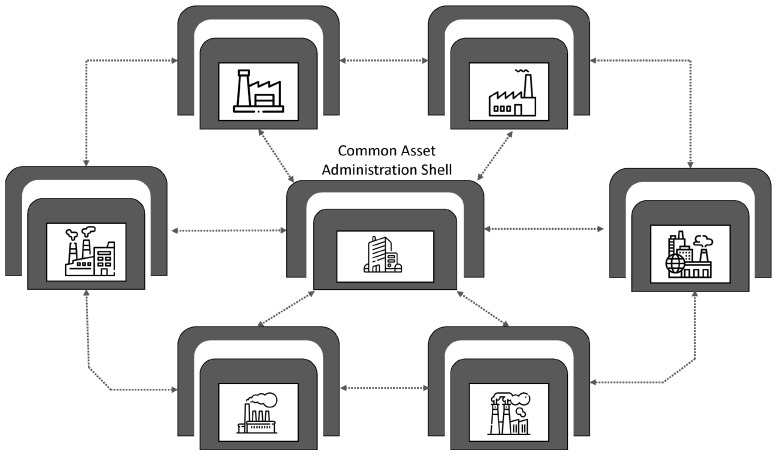
Common inter-factory (factory-to-factory) asset administration shell.

**Figure 15 sensors-22-05834-f015:**
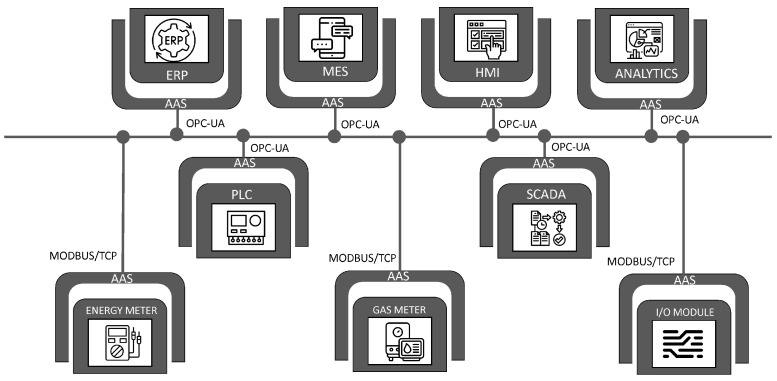
A network of assets wrapped in their AAS.

**Figure 16 sensors-22-05834-f016:**
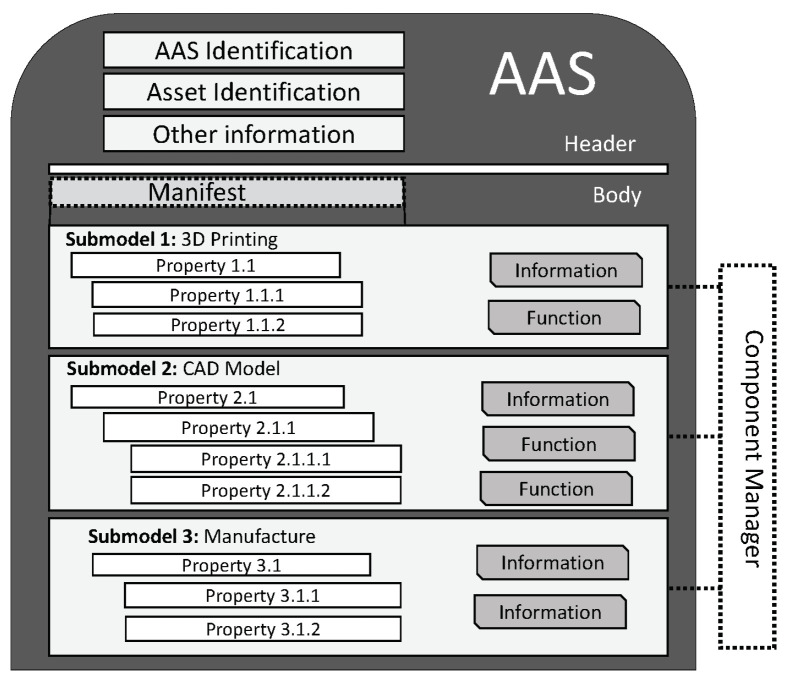
AAS metamodel for a 3D printer.

**Figure 17 sensors-22-05834-f017:**
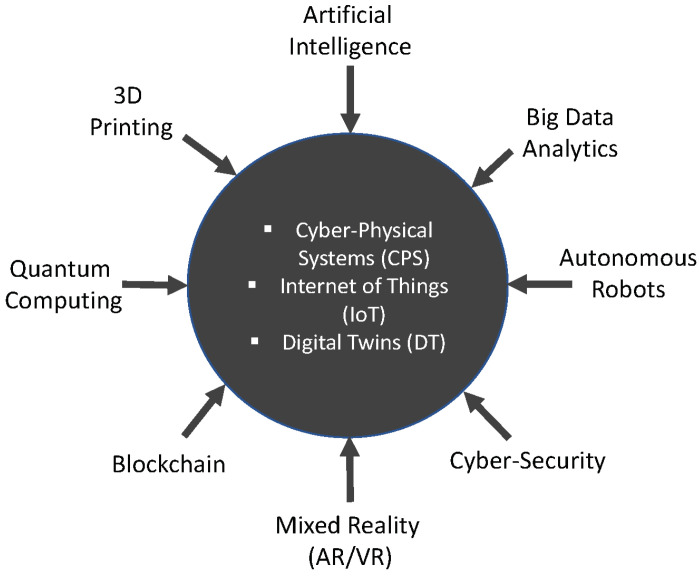
Building blocks.

**Figure 18 sensors-22-05834-f018:**
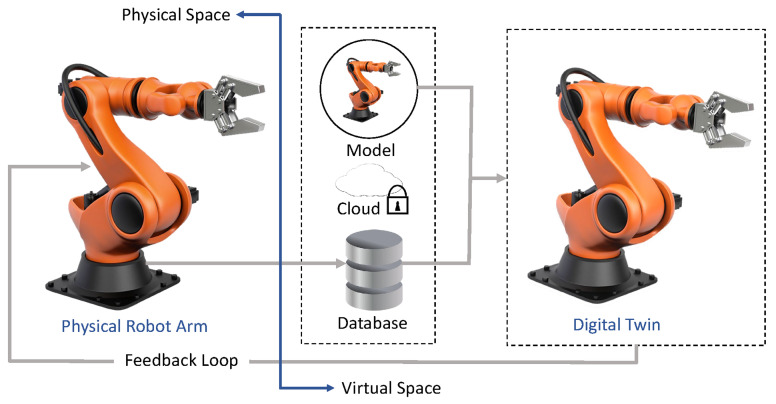
An implementation of a digital twin.

**Figure 19 sensors-22-05834-f019:**
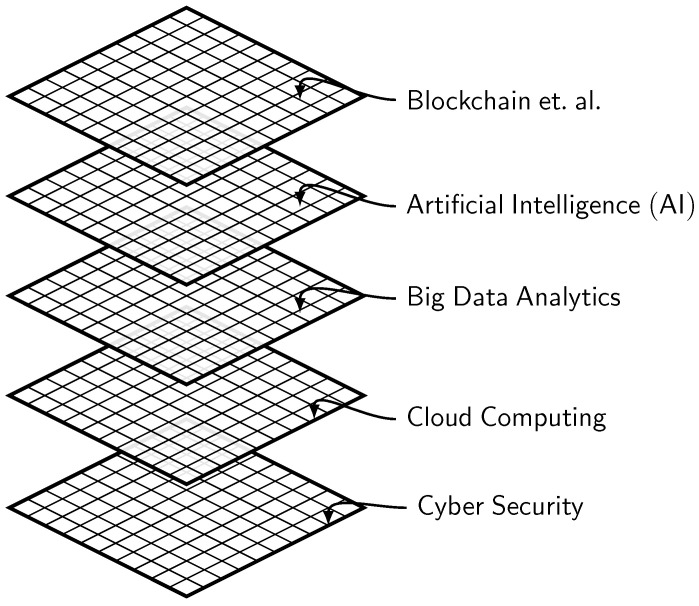
Layers of transformative technologies shaping the Future Factory.

**Figure 20 sensors-22-05834-f020:**
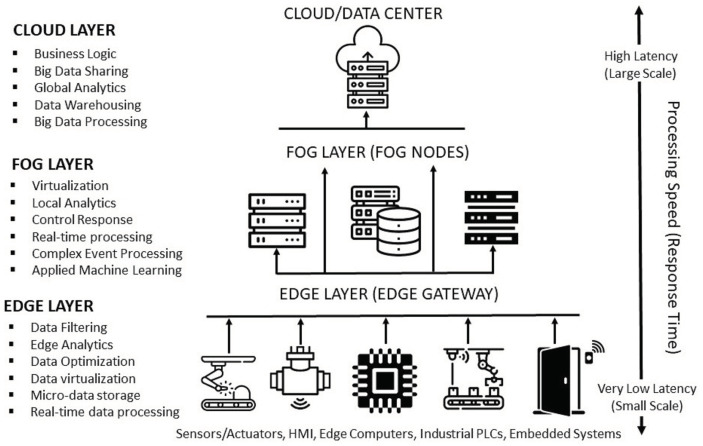
Cloud–fog–edge layer architecture.

**Figure 21 sensors-22-05834-f021:**
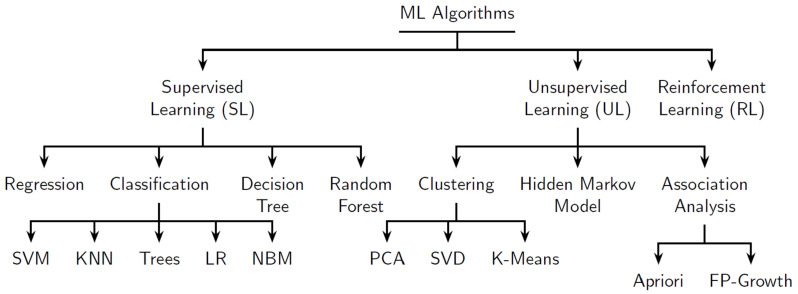
Machine learning (ML) algorithms’ *acronyms:* support vector machines (SVM), K-nearest neighbors (KNN), logistic regression (LR), Naïve Bayes multinomial (NBM), principle component analysis (PCA), singular value decomposition (SVD), frequent pattern-growth (FP-Growth).

**Figure 22 sensors-22-05834-f022:**
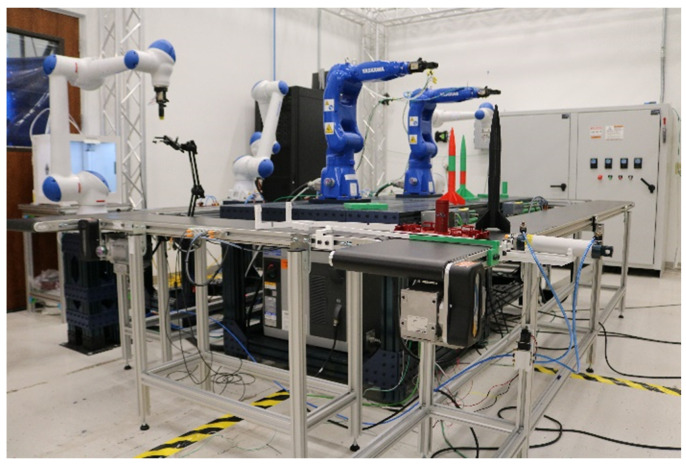
The neXt Future Factory Test Bed at University of South Carolina.

**Table 1 sensors-22-05834-t001:** Standards aligned to the ISA95 model.

ISA95 Model Levels	Tools	Standards
Enterprise Level	ERP	ISO 15704 Enterprise Architecture Requirements
		ISO 20140 Automation Systems and Integration
		ISO 19439 Enterprise Integration
		ISO 19440 Enterprise Integration
		OAGIS
		BPMN, DMN, PMML
		B2MML
MOM Level	MOM	IEC 62541, IEC 62837
		IEC 62264 (ISA 95)
		ISO 22400
		OAGIS
		PMML
		DMIS, QIF
SCADA Level	HMI/DCS	IEC 62541 (OPC UA)
		IEC 61512 (ISA 88)
		Modbus
		BatchML, PACKML
		IEC 62541 (OPC UA)
Device Level	Field Device	MT Connect
		IEC 61158 (EtherCAT, PROFINET)
		IEC 61784
		Modbus/Profibus
		PROFlenergy
		IEC 62591/HART
		IEC 62541(FDI)

## Data Availability

Not applicable.

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
