# Peer review of "A Primer on the Factories of the Future"

_sensors, 2022, doi:10.3390/s22155834_

Round 1

Reviewer 1 Report

1. I wonder if there are any other existing review papers about such a big topic. And what are the main contributions and differences between this paper and those previous publications?

2. In section 3.3.7 (Modularity), cloud computing and 3D printing are mentioned. Actually, the concept of cloud manufacturing is an extension and application of cloud computing in the manufacturing industry. Cloud manufacturing is also widely recognized as one of the possible future manufacturing modes. Therefore, it is recommended to add more discussion on cloud manufacturing and the relationship between cloud manufacturing and the factory of the future.

3. Also, there is only one line in section 3.3.7 which is too short to be a single section. It can be either merged into other sections or extended to a suitable length.

4. "PRIMER" in the title is not very suitable for a research paper. The title looks more like a book.

5. The title of Section 4 is "Conceptual Frameworks and Reference Architectures", but the two subsections of section 4 are "reference architectural model" and "Communication Standards". This is inconsistent. Should the title of section 4 be changed to "reference architectural model and Communication Standards"?

6. "neXt" in section 7 looks weird. It seems like it is the name of a lab, the "neXt Lab". It would be better to quote it like: "neXt Lab" and The "neXt Future Factory" to make it clearer.

7. Factories are very different from each other in terms of product types, production processes, automation levels, digitalization levels, and so on. So, it's hard to make a standard for all industries and all factories over the world. It's suggested to add more description of the limitations and boundaries of this review and design.

8. It's suggested to add how the authors conducted the paper searching so others can repeat the process, including the search query, time range, databases, etc.

Author Response

We would like to thank you for your thoughtful comments and efforts towards improving our manuscript. We have made attempts to respond to all the comments and would be happy to make any further adjustments, if necessary. Thank you once again.

Reviewer 2 Report

Authors prepared very interesting manuscript which provide comprehensive view to factories of the future. The paper summarized most of the currently available information on smart factories. Only minor recommendations are proposed.

Introduction

-       - Here could be mentioned the main aim of the review

-      -  Factory of future, future factory could have synonym „smart“ or „intelligent factory“

Literature background

-      -  I recommend move 1.1, 1.2, 1.3 to new chapter literature background  (in introduction then fix sections numbering)

-       - LINES 64-84 is without references. It will be good to mention some historical facts

-        -LINES 92-96. at least one  reference could be cited here

-        -Levels of system sovereignty model. Add references to semi-autonomy, autonomy chapters (IoT, CPS, PLC could refered to other works)

-        -Small headers 1.2.1,1.2.2, 1.2.3., 1.2.4. headers have symbol colon„:“ at the end, remove it

-        -Autonomy chapter – „Artificial intelligence“ could be mentioned here

Research methodology

-        -The review has semi-systematic (narritive) form. OK

-        -Divide this long paragraph in methodology into 2-3 parts.

-        -I recommend use methodology steps = first, second then …. Etc. (especially for part you mentioned keyword search, digital sources etc., describe more the proces of „search strategy“ ). Its not clear what you search and what was purpose of this search by kewords (not sure where are results of this search).

-        -lINE 263 i.e. snowballing search methodology … it is mentioned, but not depply described

Understanding the Future Factory

-        -the chapter is good prepared

-        -Capacity for mass customization (LINES 432-440) could be mentioned bass personification and referred by some publications, for example: 

o   https://doi.org/10.1007/s40436-017-0204-7

o   https://doi.org/10.3390/pr10030539

o   https://doi.org/10.1016/j.aci.2018.05.004

Conceptual Frameworks and Reference Architectures

-        -the chapter is good prepared

Realizing the promise of Industry 4.0 through the digitization of physical assets

-        -the chapter is good prepared

Key building blocks, technology enablers and innovation accelerators

-      -  This header use symbol „|“, better use classic comma „ , 

The next future factory

-       - the chapter is good prepared

Summary and Conclusion

-What you think about future research in this area, it could be more described in conclusion

Author Response

(The authors gave the same response as above.)

Reviewer 3 Report

The following corrections must be made in the article submitted for review. First of all, the abstract should clearly define the purpose, research methodology and one main conclusion. In the article, the authors did not discuss the results and did not present the limitations of their research. The authors should answer the following questions: what are they new to science? How innovative are the methods used? Why is the presented the factories of the future important from the theoretical and practical side?

The paper describes the concise review of general principles, fundamental concepts, major characteristics, key building blocks and implementation guidelines for the Future Factory within the overall context of the manufacturing ecosystem, in the age of Industry 4.0. Unfortunately, this paper has significant flaws. It is not written as a research paper but rather as a white paper or a textbook for factory engineers.

The paper is clearly written but, unfortunately, is missing in some relevant aspects, as highlighted in the following.

The theme is interesting and relevant, but the innovative aspects are fragile and must be proved and rigorously analyzed. Several issues will be listed below.

-The authors' contributions should produce measurable results that allow readers to appreciate the advancements compared with the current state-of-the-art.

-The mere professional application of a methodology does not represent an advancement of the state-of-the-art.

-The composition of the table of contents for this paper should be completely revised.

Section 1–3 partially agree, but In Section 4, reference models including Korea are newly defined, so reference models of Korea, China, Japan, etc. should be investigated and included.

In the case of Section 5, please add implementation and application examples to Sections 5.2 and 5.3.

Section 6 is very conceptually summarized and lacks research value. Each technology should be very specific about how it is implemented, where it is applied, and what its future prospects are.

In Section 7, in the case of Korea, it is necessary to investigate and summarize it for each country because it is built and operated by universities and local governments.

Based on the submitted paper, between Sections 7 and 8, please reorganize Section 8 by collecting examples of the construction of the future factory.

This reviewer is a professor who teaches smart factories in graduate schools for manufacturing innovation in Korea. This reviewer does not agree to the publication of the journal unless it is fully revised because it is only a content that has already been studied in Korea 7 years ago.

Author Response

(The authors gave the same response as above.)

Round 2

Reviewer 3 Report

I was asked to give an opinion on the revised manuscript. My recommendation after the first review was: reject, on the grounds that I do believe that the defects found in the submission can only be mended by a clean-slate approach, which questions the initial premises without being lenient on the original structure, approach and narrative.

The authors have decided to resubmit, and, as I feared, made numerous and somewhat extensive in-place additions which take for granted that what was left untouched was fine.

I agree with the authors' argument that we must document technologies necessary for manufacturing innovation. However, there are other venues where such documents can be disseminated, e.g. technical magazines and reach the right audience.

An academic journal article that publishes results based on research is structured differently than the current manuscript. Even after the revisions, I believe this paper does not describe a system that can be evaluated and replicated by the research community. 

Overall, I still believe that his paper is not ready for publication and the authors should reconsider if they want to target an academic journal or a technical magazine for disseminating their work.

Author Response

We have made an attempt to make the recommended changes. We hope the changes meet your requirements. Thank you for taking your time to go over our work. We believe your input has made the work better.  Please see attached our line by line feedback. Thank you once again. 
